

# Dynamic Consideration of Smog Chamber Experiments

Wayne K. Chuang[1] and Neil M. Donahue[1]

[1]Carnegie Mellon University Center for Atmospheric Particle Studies, Pittsburgh, USA

*Correspondence to:* Neil M. Donahue
(nmd@andrew.cmu.edu)

**Abstract.** Recent studies of the $\alpha$-pinene + ozone reaction focused on particle nucleation show relatively high molar yields of highly oxidized multifunctional organic molecules with very low saturation concentrations that can form and grow new particles on their own. On the other hand, numerous smog-chamber experiments focused on Secondary Organic Aerosol mass yields, interpreted via equilibrium partitioning theory, suggest that the vast majority of SOA from $\alpha$-pinene is semi volatile. We explore this paradox by employing a dynamical volatility basis set model that reproduces the new-particle growth rates observed in the CLOUD experiment at CERN and then modeling SOA mass yield experiments conducted at CMU. We find that the base-case simulations do over-predict observed SOA mass but by much less than an equilibrium analysis would suggest because delayed condensation of vapors suppresses the apparent mass yields early in the chamber experiments. We further find that a second model featuring substantial oligomerization of semi-volatile monomers can match the CLOUD growth rates with substantially lower SOA mass yields because the lighter monomers have a higher velocity and thus a higher condensation rate for a given mass concentration. However, we also find that if the chemical conditions in CLOUD and the CMU chamber were identical, nucleation would have occurred in the CMU experiments when in fact none occurred. This suggests that the chemical mechanisms differed in the two experiments, perhaps because the high oxidation rates in the SOA formation experiments led to rapid termination of peroxy radical chemistry.

## 1 Introduction

The mass yields of Secondary Organic Aerosols (SOA) under ambient conditions are a central issue in atmospheric chemistry. SOA production from biogenic compounds, especially monoterpenes such as $\alpha$-pinene, has been studied for decades because of its contribution to "blue haze" (Haagen- Smit, 1952) and its potentially large contribution to background aerosol concentrations both in the pre-industrial and present-day atmosphere (Kroll and Seinfeld, 2008; Hallquist et al., 2009). Traditional smog-chamber experiments have been interpreted since Odum et al. (1996) in the context of equilibrium partitioning theory (Pankow, 1994). Specifically, the mass yield of SOA in smog-chamber experiments is defined as the mass of SOA formed ($C_{OA}$) divided by the amount of precursor consumed ($\Delta C_{prec}$), measured in $\mu g\ m^{-3}$. SOA mass yields characteristically increase with increasing $C_{OA}$, and Odum's key insight was to realize that this was consistent with equilibrium partitioning theory (Pankow, 1994) and furthermore that "Odum plots" of mass yield vs $C_{OA}$ organized seemingly discordant experimental data and could be interpreted via the now widely used "two-product" equilibrium partitioning model (Odum et al., 1996).





Ozonolysis of $\alpha$-pinene has been extensively studied, and the equilibrium partitioning analysis of Odum et al. (1996) to constrain the volatility of reaction products shows a dramatic increase as the SOA loading increases. Because aerosol loading (and product volatility) can span a wide range, the SOA loading axis of the Odum plot is best expressed along a log scale (Donahue et al., 2006; Presto and Donahue, 2006). Smog-chamber experiments have typically covered a mass concentration range of $1 \lesssim C_{\mathrm{OA}} \lesssim 1000\,\mu\mathrm{g\,m^{-3}}$, with and without inorganic seeds to promote condensation of vapors (Odum et al., 1996; Griffin et al., 1999; Cocker III et al., 2001; Presto et al., 2005; Presto and Donahue, 2006; Pathak et al., 2007a; Shilling et al., 2008, 2009; Song et al., 2007). These data show little to no mass yield for $C_{\mathrm{OA}} \lesssim 1\,\mu\mathrm{g\,m^{-3}}$, but for $1 \leq C_{\mathrm{OA}} \leq 1000\,\mu\mathrm{g\,m^{-3}}$, the mass yield increases dramatically. Even studies with high seed surface area (Pathak et al., 2007b; Song et al., 2007) and continuous-flow chambers that should encourage equilibration (Shilling et al., 2009) show mass yields below 10% at low $C_{\mathrm{OA}}$, though the results of Song et al. (2007) and Shilling et al. (2009) approach 10%. The equilibrium partitioning analysis relates the volatility of an organic species to determine at which loading ($C_{\mathrm{OA}}$) a compound would contribute significantly to the SOA mass yield (Presto and Donahue, 2006). For instance, for an aerosol loading of $C_{\mathrm{OA}} = 10\,\mu\mathrm{g\,m^{-3}}$, an organic species with a volatility of $C^* = 10\,\mu\mathrm{g\,m^{-3}}$ will partition 50% into the gas phase and 50% into the (organic) particle phase. If the loading were 10 times lower, at $C_{\mathrm{OA}} = 1\,\mu\mathrm{g\,m^{-3}}$, the species would partition 90% into the gas phase and 10% into the particle phase. This equilibrium behavior motivates the volatility basis set (VBS), which separates compounds into volatility bins, each an order of magnitude apart (Donahue et al., 2006). In this way, an equilibrium partitioning analysis of smog-chamber data for $\alpha$-pinene SOA parses the yield data to form a distribution of compounds by their volatilities, with few to no compounds with low volatilities, $C^* \lesssim 1\,\mu\mathrm{g\,m^{-3}}$, and most of the mass with volatilities $1 < C^* < 10^6\,\mu\mathrm{g\,m^{-3}}$ (Presto and Donahue, 2006).

Recent experiments using nitrate-ion clustering chemical ionization mass spectrometry (nitrate CIMS) have revealed the presence of "highly oxidized multifunctional organics" (HOMs) that have been interpreted as Extremely Low Volatility Organic Compounds (ELVOCs) and Low Volatility Organic Compounds (LVOCs, collectively (E)LVOCs) in the VBS nomenclature (Ehn et al., 2014). The molar yield of HOMs was initially estimated to be $7 \pm 3.5\%$ (Ehn et al., 2014), and their volatilities are thought to be much lower than $10^{-1}\,\mu\mathrm{g\,m^{-3}}$ based on their molecular formulas and assumed structures. Recent experiments conducted in the CLOUD chamber at CERN confirmed a wide distribution of HOMs from oxidation of $\alpha$-pinene, especially by ozone, with the estimated volatility ranging from $10^{-20} \lesssim C^* \lesssim 10^{-2}\,\mu\mathrm{g\,m^{-3}}$ (Tröstl et al., 2016; Kirkby et al., 2016). The nominal molar yields from the CLOUD nitrate-CIMS measurements based on sulfuric acid calibrations were at the low end of prior measurements, near 3.5% (Kirkby et al., 2016), but flux-balance calculations based on the observed particle growth rates require more than three times the mass flux than can be explained by those nominal values. Because the nitrate CIMS relies on clustering between polar functional groups and the nitrate anion, which broadly corresponds with what makes the compounds have a low vapor pressure and stick to small particles, and because more volatile species that dominate ($> 90\%$) the molar product distribution are invisible to the nitrate CIMS, Tröstl et al. (2016) proposed that the clustering efficiency of the nitrate CIMS scales with $C^*$ and that the efficiency drops off in the LVOC volatility range. LVOC yields based on this empirically derived clustering efficiency quantitatively explain the observed particle growth rates for $1 \leq d_p \leq 30$ nm, both at constant measured HOM concentrations and when the HOM concentrations are rising steadily. However, with molar yields well over 10% the derived mass yields of these highly functionalized ELVOC and LVOC products exceeds 30%.



The high mass yields of (E)LVOC products based on direct CIMS measurements from Ehn et al. (2014) and both CIMS measurements and dynamic flux balances based on growth rates from Tröstl et al. (2016) appear to contradict the earlier smog-chamber studies of $\alpha$-pinene ozonolysis SOA mass yields. We illustrate this in Figure 1, where we contrast the VBS equilibrium partitioning analysis carried out by Presto and Donahue (2006) with the equilibrium expectations of the nitrate-

5   CIMS (E)LVOC observations. In the equilibrium analysis we expect rising mass yields where $C^* \simeq C_{\mathrm{OA}}$, indicated by the stacked histogram showing 50% partitioning for bins with $C^* = C_{\mathrm{OA}}$ and the black equilibrium partitioning curve. In contrast, if the mass yield of (E)LVOCs is of order 30% and even if they are at the extreme high end of the LVOC range suitable for condensation in Tröstl et al. (2016) with $C^* \simeq 10^{-2}\,\mu\mathrm{g}\,\mathrm{m}^{-3}$, we would expect the observed mass yields to rise to 30% by the time $C_{\mathrm{OA}} \geq 10^{-1}\,\mu\mathrm{g}\,\mathrm{m}^{-3}$ in an ideal, loss-free chamber, as shown by the solid green curve. There is a vast difference between

10   the two curves. If these (E)LVOC products exist at such high mass yields, the simple question is thus: why do they not appear as high mass yields at low $C_{\mathrm{OA}}$ in the Odum plots from SOA experiments? There are several possibilities:

1. Dynamical effects could delay condensation and thus bias the observed mass yields low for a given amount of precursor loss,

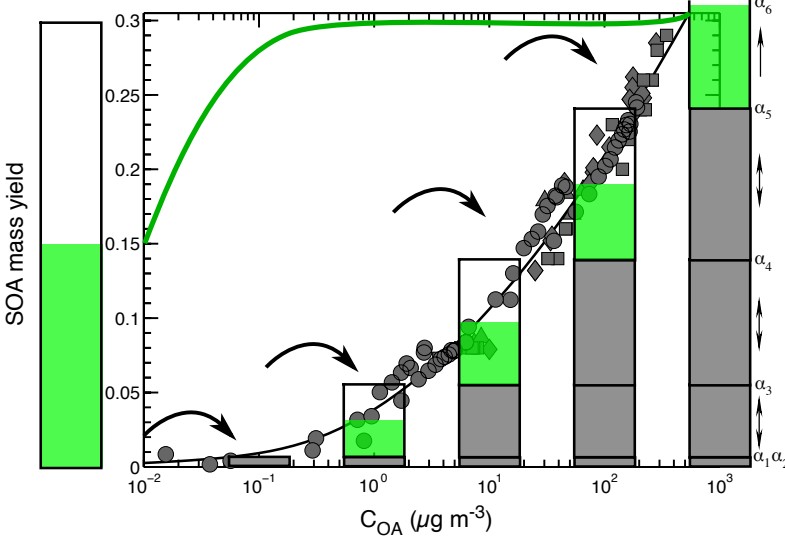

**Figure 1. Apparent contradiction between CIMS (E)LVOC measurements and chamber SOA mass yields.** A VBS equilibrium partitioning analysis for $\alpha$-pinene ozonolysis SOA compared with equilibrium partitioning expected from CIMS observations showing high mass yields of extremely low volatility (E)LVOC products. Vertical bars represent mass yields in volatility bins, with filled gray representing species with $C^* \ll C_{\mathrm{OA}}$ and green under clear representing 50:50 partitioning of species with $C^* \simeq C_{\mathrm{OA}}$. Data and an equilibrium VBS analysis are shown within the frame, while nominal equilibrium partitioning for a single LVOC constituent with a mass yield of 0.3 and $C^* = 0.01\,\mu\mathrm{g}\,\mathrm{m}^{-3}$ is shown with the offset bar and the green curve. The gap between the black and green curves represents the paradox motivating this paper.





2. Condensible vapor loss to the chamber walls could bias the observed mass yields low,

3. Oligomerization and not direct condensation of (E)LVOCs could explain some of the CLOUD growth-rate observations, with uncertain effects on the SOA chamber mass yields,

4. High oxidation rates in the SOA chambers could interfere with HOM formation via the peroxy-radical auto-oxidation mechanism.

In the equilibrium partitioning interpretation, HOMs would instantly condense into the particle phase and show a high mass yield at low aerosol loading. However, barriers to condensation such as the Kelvin effect, a low mass accommodation coefficient, or simply a low aerosol surface area, could delay the effects enough that this mass does not appear until more $\alpha$-pinene has reacted, thus lowering the observed mass yield. Further, if condensation to suspended particles is low, vapor wall losses may be high (Matsunaga and Ziemann, 2010; Ye et al., 2016a; Trump et al., 2016). While the growth-rate data demand that the eventual reaction products have a very low volatility, it is possible that condensed-phase chemistry ("oligomerization") (Kalberer et al., 2004; Tolocka et al., 2004) could produce ELVOC products in the CLOUD chamber on a timescale of several hours, driving the 2-6 nm hr$^{-1}$ growth rates, but be less evident in chamber SOA experiments where typical conditions involve $\alpha$-pinene oxidation in well under an hour and growth rates above 100 nm hr$^{-1}$. Finally, those high oxidation rates in the chamber experiments could interfere with the RO$_2$ auto-oxidation chemistry (Crounse et al., 2013; Ehn et al., 2014; Rissanen et al., 2014) by shortening the RO$_2$ bimolecular lifetime, thus sharply reducing the (E)LVOC mass yields in the SOA chamber experiments.

In this study, we begin by modeling aerosol growth dynamically within a VBS framework. Our objective is to explore whether the mass yields required to explain the growth rates observed in CLOUD do indeed over-predict the SOA chamber observations, as suggested by Figure 1, or whether some combination of dynamics, wall losses, and condensed-phase chemistry may reconcile this apparent contradiction. Because of this, we shall consider *only* condensible products required to explain the CLOUD growth-rate observations (consisting broadly of products with $C^* \leq 10^{-2}\,\mu\mathrm{g\ m^{-3}}$, whether formed in the gas or the condensed phase). We shall correct for the temperature difference, as the CLOUD experiments were conducted at 278 K and typical chamber SOA experiments have been conducted almost 20 K higher in temperature (corresponding to approximately a one-decade shift in volatility toward higher values in the SOA chamber experiments). Besides that, however, we shall not model production or condensation of any SVOC products (other than reactive monomers that ultimately oligomerize). The question is whether this reduced set of (E)LVOC products over-predicts SOA chamber mass yield experiments; any under-prediction would presumably be due to condensation of SVOCs in those experiments.

Recent studies imply that a dynamic approach is necessary to capture the interactions between the organics in the vapor phase and the suspended phase (McVay et al., 2014, 2016), and the loss of vapors and particles to the chamber walls (Zhang et al., 2014). Because condensation is not instantaneous, some condensible vapors are lost to the walls instead of settling onto particles. The dynamic model accounts for the time it takes for vapors to interact with particles and condense, or hit the chamber wall and become absorbed by the Teflon.





We can use the model to explore how changes in chamber experiment parameters can change the production (mass yields) of organic aerosols. The production of particle mass depends on the ratio of the particle condensation sink to the wall loss sink. The particle condensation sink scales approximately proportionally with particle surface area. Therefore, to decrease the wall loss of condensible vapors, chamber experiments often use ammonium sulfate seeds to encourage condensation as opposed to relying on nucleation, which can result in high wall loss of condensible vapors early in experiments when the nucleated particle condensation sink is very low (Kroll and Seinfeld, 2005). However, the polydisperse seeds generated often span a wide size range, over an order of magnitude or more. At any point in time, each particle size has a different condensation sink, which affects the growth rate of the particle. This complicates calculations, as each particle would have a different growth rate and also (transiently) a different composition due to different surface-area to volume ratios. The polydispersity may also have implications in particle-phase chemistry (Shiraiwa et al., 2013), though that is not explored here. Saleh et al. (2013) showed that it is possible to use a monodisperse population with the size of a condensation sink diameter to approximate the dynamical behavior of a polydisperse aerosol suspension. We utilize a condensation sink diameter to compare the polydisperse and monodisperse versions of the model, and confirm that the condensation sink diameter provides a good approximation. We also look at how changes in the ratio between the particle condensation sink and the vapor wall loss affects production of suspended organic aerosol.

## 2 The Dynamic Model

### 2.1 Mathematical background

We modeled the production of $\alpha$-pinene SOA using a dynamic 1-dimensional VBS, meaning that we treat volatility only and not the composition of the organics. This was previously discussed in the supplemental material for Tröstl et al. (2016), but here we summarize the essential features. The VBS product distribution spans a volatility range $10^{-8} \le C^* \le 10^{-1}\,\mu\text{g m}^{-3}$, covering extremely-low-volatile to semi-volatile organic compounds (ELVOCs and LVOCs).

Interactions between the bulk vapors and suspended particles, and between chamber walls, are described by a set of ODEs for each volatility bin $i$:

$$\frac{dC_i^v}{dt} = P_i^{\text{prec}} - \phi_i^{v,s} - \phi_i^{v,t} \tag{1}$$

$$\frac{dC_i^s}{dt} = \phi_i^{v,s} - \phi_i^{s,d} \tag{2}$$

$$\frac{dC_i^t}{dt} = \phi_i^{v,t} \tag{3}$$

$$\frac{dC_i^d}{dt} = \phi_i^{s,d} \tag{4}$$

where superscripts identify reservoirs: $v$ is vapor; $s$, suspended particles; $t$, teflon(wall)-absorbed vapors; and $d$, wall-deposited particles. The superscript order is a transfer of mass from the first to the second reservoir. $P_i^{\text{prec}}$ is the production of vapors through $\alpha$-pinene ozonolysis, and is distributed according to the mass yield for each VBS bin. Vapor-phase HOMs generated





through oxidation of $\alpha$-pinene either condense onto suspended particles ($\phi_i^{v,s}$) or are irreversibly lost to the walls ($\phi_i^{v,t}$). Vapor wall loss is a first-order loss rate $\phi_i^{v,t} = k^{v,t} C_i^v$ with a timescale of 10 minutes (Ye et al., 2016a; Krechmer et al., 2016; Trump et al., 2016). This is currently assumed to be irreversible, due to the low volatility of the HOMs and the high effective saturation concentration of the walls (McVay et al., 2016). A major difference between the CLOUD experiment

and SOA production experiments in teflon chambers is that in CLOUD the collision frequency (condensation sink) of vapors to the walls typically exceeds the suspended condensation sink, whereas in most chamber SOA experiments the suspended condensation sink exceeds the wall collision frequency. Also, CLOUD is stainless steel whereas most SOA smog chambers are Teflon; especially on the metal surfaces, it is possible that reactive uptake (i.e. decarboxylation) is important. However, in each case we model the vapor wall loss as irreversible. We do not treat reversible sorption to the Teflon in this work as

proposed by Matsunaga and Ziemann (2010) because our objective is to identify the maximum possible wall interference and in any event the (E)LVOCs have a very low equilibrium vapor concentration over the walls (Krechmer et al., 2016). Organics in the suspended phase can evaporate into the bulk vapor ($-\phi_i^{v,s}$); alternatively, the particle itself with its mix of organics and seed can be irreversibly lost to the walls. This is determined by data on the first order loss rate of SOA in the chamber ($\phi_i^{s,d} = k^{s,d} C_i^s$).

In a typical experiment, ammonium-sulfate seeds are first injected into a cleaned empty chamber to provide a condensation sink and also to constrain the particle wall loss rate constant. Then $\alpha$-pinene and ozone are added, producing HOMs that condense to the walls or seeds. For the experiments we explicitly model here, the suspended particle evolution was monitored with a Scanning Mobility Particle Sizer (SMPS), which measures particle volume but does not differentiate between organics and seeds. In order to separate the two, we rely on the seed loss rate measured prior to the injection of $\alpha$-pinene and extrapolate

the seed concentration subsequent to the injection. This results in minor discrepancies between the data and the model concerning the mass of seeds in the chamber, but does not have a major effect on our overall conclusions. Because of the many time-dependent influences, such as wall losses and delays to condensation, we shall focus on directly comparing measured to modeled suspended-particle mass (i.e. without any wall loss corrections) to determine whether the CLOUD constrained products over- or under-predict the chamber SOA results.

## 2.2  (E)LVOC mass yield distribution

Tröstl et al. (2016) showed that (E)LVOC species observed in the CLOUD experiment at CERN could explain observed growth rates in experiments where particle nucleation and growth was driven exclusively by $\alpha$-pinene ozonolysis. However, the observed growth was substantially faster than the raw nitrate-CIMS measurements could explain, and so those authors hypothesized that LVOC species are inefficiently detected by the nitrate clustering and thus that the actual LVOC concentrations

in CLOUD were significantly higher than the nominal concentrations (which are based on a sulfuric acid calibration). Here we shall retain this interpretation as our base case. However, the CLOUD chamber experiments were conducted at 278 K, we wish to apply those results to CMU smog-chamber data collected near room temperature, and volatility depends on temperature. By applying the Clausius-Clapeyron equation and assuming an enthalpy of vaporization of 110 kJ mol$^{-1}$ (Bilde and Pandis, 2001; Sheehan and Bowman, 2001; Epstein et al., 2010), we estimate that an increase of 15 K results in approximately one order of





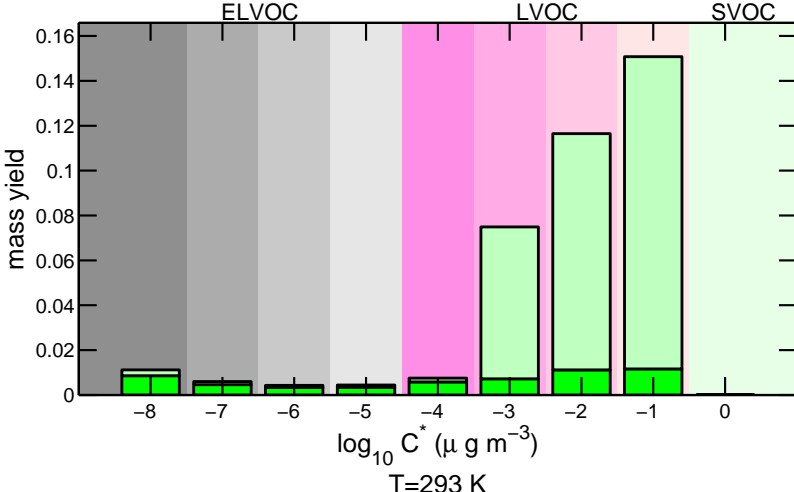

**Figure 2. Distribution of Highly Oxidized Multifunctional Organic molecules from $\alpha$-pinene + ozone.** Mass yields are consistent with product and growth-rate observations from CLOUD, but adjusted to 293 K consistent with typical Secondary Organic Aerosol chamber experiments. The dark green bars are the raw mass yields detected by the nitrate-CIMS. The light green bars show additional mass required to reproduce CLOUD growth rates, assuming that the nitrate clustering efficiency in the CIMS declines with increasing product volatility.

magnitude increase in volatility. The resulting mass-yield distribution, shown in Figure 2, is {.011, .0060, .0043, .0044, .0075, .075, .12, .15} for volatility bins $10^{-8} \leq C^* \leq 10^{-1}$ $\mu$g m$^{-3}$. The dark green portion of the bars corresponds to the mass yields based on nominal nitrate-CIMS measurements, while the light green portion is the additional concentration required to explain the observed growth rates in CLOUD. This distribution (at 278 K) reproduces the particle growth rates for two

5  different experimental conditions in CLOUD, as discussed in Tröstl et al. (2016). However, it does *not* conserve carbon. The total mass yields are roughly 0.38 and the corresponding molar carbon yields are 0.23, so the distribution explains roughly one quarter of the ozonoylsis products. The remaining products, with a molar yield of 0.77, are too volatile to cause condensational growth under the CLOUD conditions. Those include SVOC products that may well condense in chamber experiments. There is compelling evidence that between 30% and 60% of the SOA from $\alpha$-pinene ozonolysis behaves like SVOCs, either evaporating

10  during dilution (Grieshop et al., 2007; Vaden et al., 2011) or transferring between different suspended populations when they are mixed (Robinson et al., 2013; Ye et al., 2016b). However, our objective is to test whether the (E)LVOC products alone pose a mass-balance problem for the chamber SOA experiments, and so in the following simulations we shall completely neglect any SVOC production, instead leaving any potential gaps between the (E)LVOC condensation and the total observed SOA to be explained by SVOCs.





### 2.3 Polydispersity and the condensation sink diameter

The condensation sink of vapors to particles is dependent on total available surface area, and thus the size and number concentration of the seed particles. As shown previously in Tröstl et al. (2016), the condensation flux is defined as:

$$\phi_{i,p}^{v,s} = N_p \underbrace{(\pi/4(d_p + d_i)^2)}_{\substack{\text{particle-vapor} \\ \text{collision cross-section,} \\ \sigma_{v,p}}} \overbrace{\alpha_{i,p} v_{i,p} B_{i,p}}^{\substack{s_{i,p}, \\ \text{deposition rate of} \\ \text{vapors at the surface}}} \underbrace{[C_i^v - a_{i,p}' C_i^\circ]}_{\substack{\text{driving force of} \\ \text{condensation,} \\ F_{i,p}}} \tag{5}$$

where $N_p$ is the particle number concentration of a specific particle type (size or composition), $d_p$ is the particle diameter, $d_i$ is the effective spherical diameter of molecule $i$ in the vapor phase, $\alpha_{i,p}$ is the accommodation coefficient, $C_i^v$ is the vapor concentration, $a_{i,p}'$ is the activity of the organics in the particle phase, and $C_i^\circ$ is the saturation vapor concentration over a pure, flat, sub-cooled liquid surface. The total condensation flux is the sum over all particle sizes and types: $\phi_i^{v,s} = \sum_p \phi_{i,p}^{v,s}$.

However, particles in chamber studies are rarely monodisperse; they often vary in size by up to an order of magnitude. When
simulating the condensed-phase composition it is computationally more efficient and conceptually more straightforward to consider a monodisperse distribution. A polydisperse model can be approximated by a monodisperse model using a condensation sink-weighted average diameter to represent the total particle population with the appropriate vapor-particle equilibration timescales. The condensation sink diameter is the diameter that monodisperse particles would have to preserve the condensation sink and the total number concentration of a polydisperse population. This does *not* conserve the seed-particle mass (it
roughly conserves surface area), so the seed mass concentrations in these simulations does not match observations. We determine the condensation-sink diameter by summing the contribution to the condensation sink from each size bin, and calculating the diameter of a monodisperse seed that would produce the same condensation sink. In other words, we find a monodisperse seed of size $d_p^{\text{CS}}$ such that:

$$k_c(d_p^{\text{CS}}, \sum_j^n N_{p,j}) = \sum_j^n k_{c,j}(d_p, N_p) \tag{6}$$

where

$$k_c(d_p, N_p) = N_p(\pi/4(d_p + d_i)^2)\alpha_{i,p} v_{i,p} B_{i,p} \tag{7}$$

In the following simulations we compare simplified cases with a monodisperse seed population set initially at the seed condensation sink diameter with a polydisperse simulation in which we initialize the simulation using the seed size distribution spread over 108 distinct particle sizes, and then allow the diameter of each seed bin to evolve as net condensation dictates.



## 3   Results and Discussion

To compare the CLOUD (E)LVOC mass yields with smog-chamber experiments, we simulate data from two experiments described by Pathak et al. (2007b), both of which had relatively high initial seed surface area and thus should have had low (E)LVOC particle wall loss and rapid equilibration. Both experiments were conducted near room temperature, common in many other smog-chamber experiments. Experiment 1 was conducted with 17 ppb $\alpha$-pinene, a constant 250 ppb $O_3$, and 12000 $cm^{-3}$ ammonium-sulfate seeds. Experiment 2 was conducted with 38.3 ppb $\alpha$-pinene, a constant 250 ppb $O_3$, and 6000 $cm^{-3}$ ammonium-sulfate seeds. The SMPS data for these experiments show clear volume maxima after SOA condensation as well as periods where particle wall losses clearly dominate; these are essential to constrain the model. The data also show a steady decline in total particle number with no sign of nucleation after the onset $\alpha$-pinene ozonolysis.

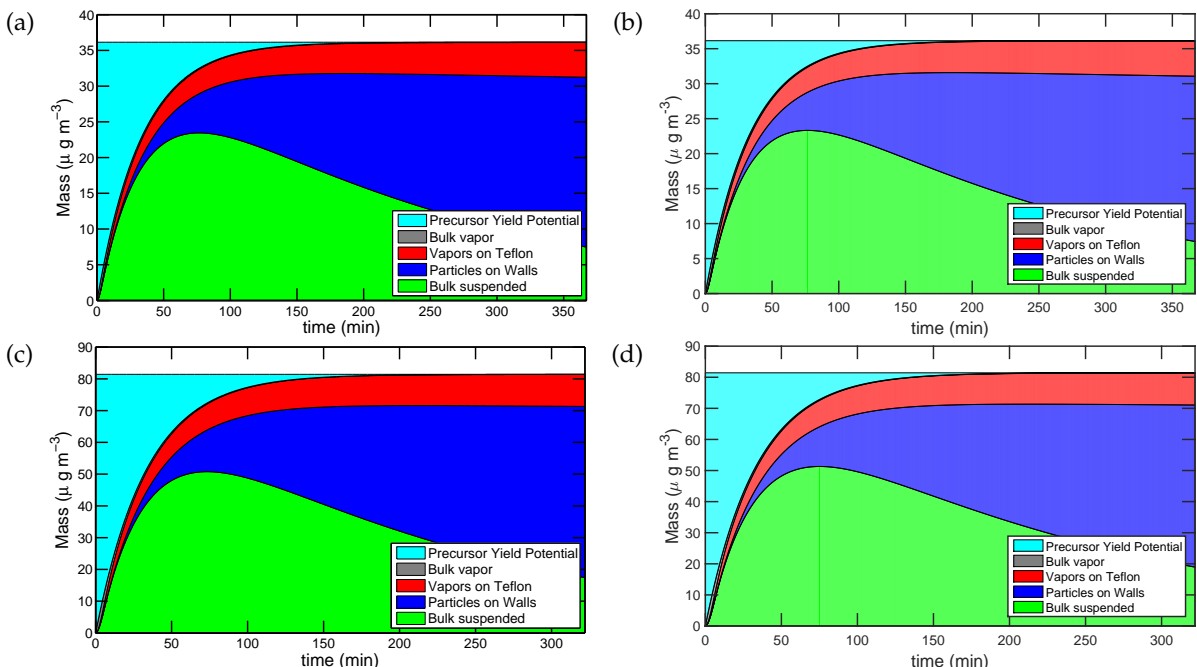

**Figure 3. Dynamical simulation of $\alpha$-pinene SOA for two experimental conditions, using a monodisperse and polydisperse model.** The simulations describe five different reservoirs: unreacted precursor, vapors, suspended particles, deposited particles, and sorption to teflon, as shown in the legend. Experiment 1 monodisperse **(a)** and polydisperse **(b)** results. Experiment 2 monodisperse **(c)** and polydisperse **(d)** results. The monodisperse model uses a "condensation sink diameter" to approximate the rate that organics condense onto particles. This serves as a good proxy for a polydisperse model that accounts for the different condensation sinks for a polydisperse seed distribution.



### 3.1 Modeling organic aerosol production

In Figure 3 we show show simulations of the two experiments. In Experiment 1 the $\alpha$-pinene oxidation produces a total of about $36\,\mu\mathrm{g}\ \mathrm{m}^{-3}$ of (E)LVOC products, while in Experiment 2 the oxidation produces about $81\,\mu\mathrm{g}\ \mathrm{m}^{-3}$ of (E)LVOC products. Because the products are effectively non volatile and the seed concentrations were similar, aside from scaled y

axes the simulations look very similar. Figure 3a and Figure 3c are results from the monodisperse model using the weighted condensation-sink diameter, and Figure 3b and Figure 3d are the polydisperse model results. The different colors denote different reservoirs of organics. The light blue is the concentration of organics that have not yet been formed by ozonolysis– essentially a proxy for the $\alpha$-pinene remaining. The grey, which will be shown more prominently later, is the oxidized products that are in the vapor phase, $C^v$ – these are products that have yet to condense. The red is vapors that have been absorbed into

the Teflon walls of the chamber, $C^t$. The dark blue is organics condensed onto particles that subsequently were deposited to the chamber walls, $C^d$. The green is organics that have condensed but remain suspended in the bulk of the chamber, $C^s$.

There is very little difference between the monodisperse (condensation sink diameter) and polydisperse models, which is immediately evident upon inspection of Figure 3. Furthermore, the large majority of the condensible mass condenses onto suspended particles that are then lost to the chamber walls (the green and blue swaths). At this scale condensible vapors still in

the gas phase (gray) appear to play a minor role. Relatively little mass (the red swath) condenses directly to the chamber walls, and so a back extrapolation of the suspended particle mass to $t = 0$ would result in a reasonably accurate estimation of the total SOA mass yield at the end of the experiment.

In Figure 4 we compare the chamber aerosol mass data and the model results for both the suspended seed mass concentration and the suspended organic aerosol mass concentration over the duration of each experiment. Here the organic aerosol is shown

in green and the ammonium-sulfate seed mass in hashed red. In both cases, the model substantially over-predicts the observed organic mass concentrations at all times, but the mismatch is significantly greater for Experiment 1, which also had less than

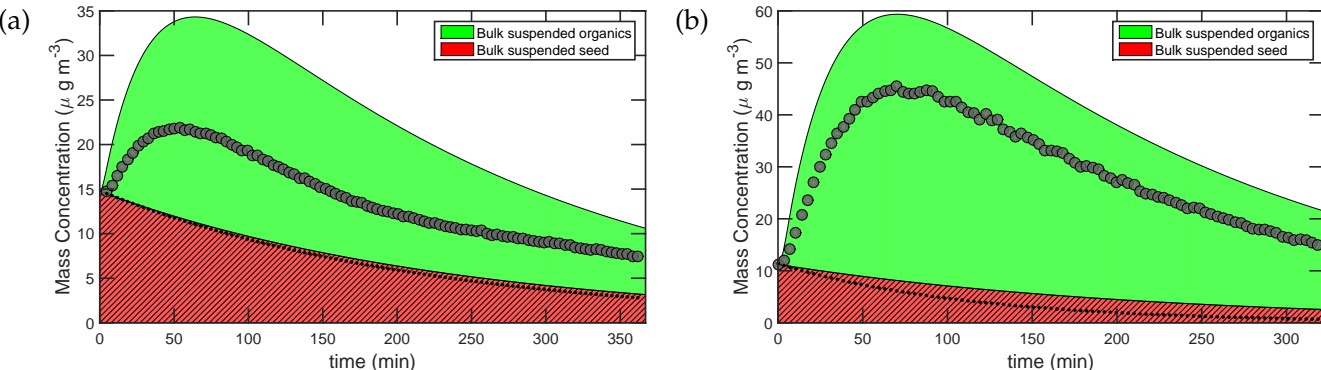

**Figure 4. Comparison of the CLOUD-constrained VBS model and the CMU chamber experiments for $\alpha$-pinene ozonolysis.** For both Experiment 1 **(a)** and Experiment 2 **(b)** the data (symbols) lie well below the model results of seed mass and organic mass over the full course of the experiment. This may be due to different experimental conditions, primarily the higher rate of ozonolysis in the SOA experiments.





half the total aerosol mass loading. Because the model treats only (E)LVOC formation and omits any SVOCs, this strongly suggests that there is a discrepancy between the mass yields required to explain particle growth rates in CLOUD and the mass yields observed at much higher concentrations in smog-chamber studies. Delayed condensation and wall losses of (E)LVOC vapors are likely not a sufficient explanation for the disagreement, as both are treated in the model. However, the larger

discrepancy at lower mass loading is an example of the type of data that inspired the Odum et al. (1996) interpretation; observed SOA mass yields tend to be lower at lower mass loading, and the reference point here is an essentially non-volatile suite of (E)LVOCs.

In Figure 5 we show the Odum plots – the mass yield of SOA versus the total organic mass produced – for both simulations. In this case we assume a perfect correction for the deposited particle mass, and so with the solid green curve we plot the total

10 SOA concentration at any given time: $C_{\mathrm{OA}} = C^s + C^d$. However, the mass yield is given by $\Delta\alpha$-pinene$/C_{\mathrm{OA}}$ and so excludes any vapors yet to condense as well as any vapors lost to the teflon walls. With the dashed curve we show the equilibrium partitioning, which is the expected mass yield if the system were to reach instantaneous equilibrium without any vapor wall losses, as depicted in Figure 1. Even though there is little evident vapor in Figure 3, here we see that there is a dramatic difference between the dynamic and equilibrium cases. This is because the difference is confined to relatively small $C_{\mathrm{OA}}$

values early in the run, and they simply fail to register on the linear scale of Figure 3. There is also a small difference in the asymptotic values of the dynamical and equilibrium models because of the vapor wall losses, but this is less significant than the dynamical delay of condensation.

We plot various SOA mass yields presented in the literature in Figure 5. In red we plot the time-dependent mass yields from Pathak et al. (2007a), including Experiments 1 and 2. The red data and green model curves are thus directly comparable. The

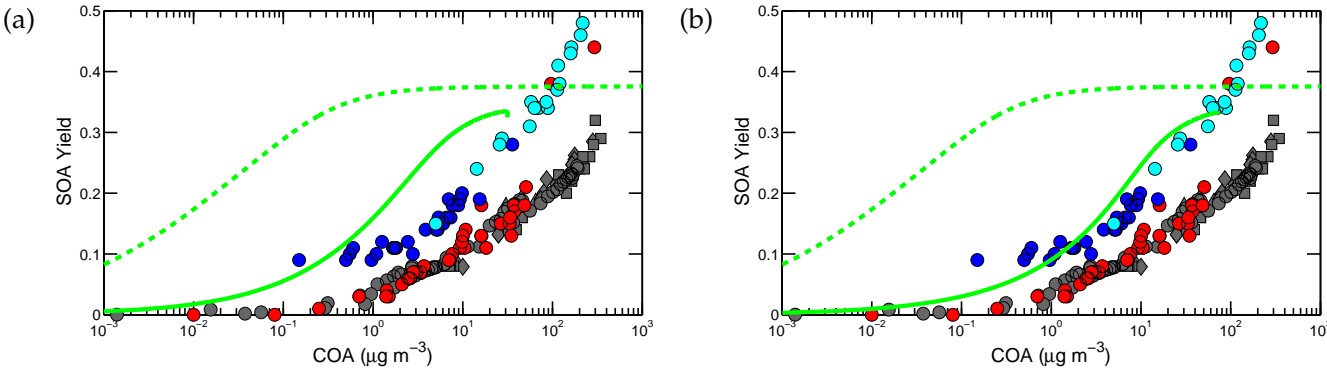

**Figure 5. Odum plots from model runs of the experiments.** Model results for Experiment 1 **(a)** and Experiment 2 **(b)** as solid green curves, compared to prior data from Presto and Donahue (2006) in grey, Shilling et al. (2008) in blue, Pathak et al. (2007a) in red, and Song et al. (2007) in cyan. The Odum plots show the model predicting higher yields than prior experiments. The dotted green curve is the equilibrium partitioning yield at a certain total organic aerosol mass. The model demonstrates that there is a significant time delay to condensation, as the solid line is far below the equilibrium line. Thus, it is possible to have substantial production of low volatility products from $\alpha$-pinene ozonolysis that results in the experimental data shown in this plot.



solid green dynamical simulation and the red data points disagree, consistent with the overshoot also evident in Figure 4. In gray we show yields discussed in Presto and Donahue (2006), which include chamber results from Odum et al. (1996), Griffin et al. (1999), and Cocker III et al. (2001). Those data agree well with the cases we are modeling here.

In Figure 5 we also also show SOA mass yields from Shilling et al. (2008), and Song et al. (2007), which are significantly

higher than the (older) data. These experiments agree reasonably well with the current simulations, though for most of the data from Shilling et al. (2008) the chamber was operated in a CSTR mode, and our simulations are for a batch mode, so the comparison should be made with care. However, the different mass yields reported by Shilling et al. (2008), and Song et al. (2007) raise the possibility that different experimental conditions in different chamber studies might partially explain the apparent discrepancy between mass yield and growth-rate observations.

The difference between the dynamical and equilibrium simulations evident in Figure 5 shows that one cannot necessarily assume equilibrium partitioning when determining SOA mass yields. The model demonstrates that when smog chambers are not treated dynamically, it is possible to miss substantial yields of low volatility organic compounds that are effectively held up in the gas phase before condensing. However, the simulations still predict substantially more SOA at any given $C_{OA}$ than we have reported previously, so this delay does not by itself resolve the apparent discrepancy.

Part of the dynamical effect is the delay between the production and condensation of (E)LVOCs. We show this delay more clearly in Figure 6a by focusing on the first few minutes of Experiment 1, shown in Figure 3a. As the experiment starts, the amount of oxidized $\alpha$-pinene increases nearly linearly, but the bulk vapor concentration (grey) grows substantially before condensation to the bulk suspended particles begins to be significant. In addition, this reservoir remains as the experiment progresses because there is always a steady-state concentration of condensible vapors driving particle growth, indicating that

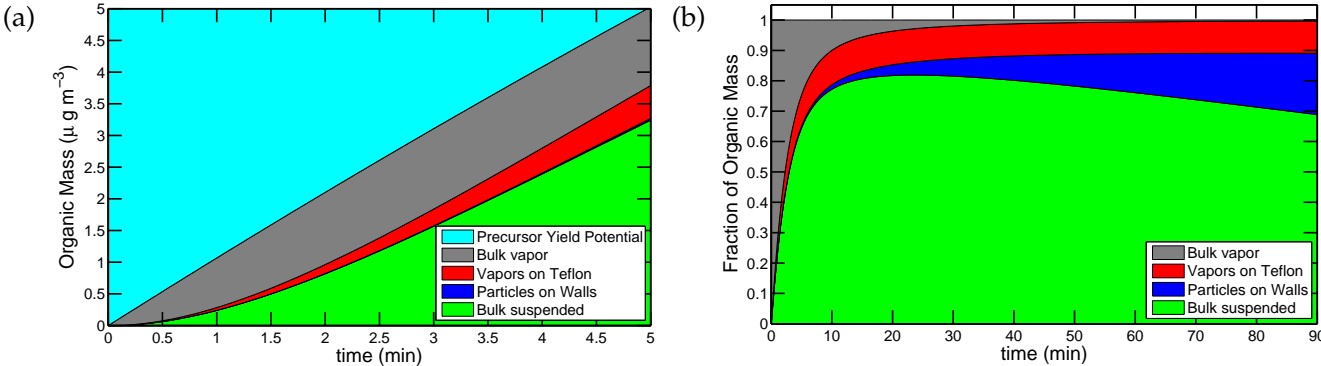

**Figure 6. Mass distribution among organic reservoirs.** The first five minutes of Experiment 1 **(a)** show the different reservoirs of organic mass including a substantial fraction of uncondensed vapors (gray). The buildup of the bulk vapors demonstrates that there is a significant delay between the formation of low volatility compounds and the condensation of these compounds onto particles. This results in lower detected yields during chamber experiments and the loss of vapors to the walls. The fraction of organics in each of the reservoirs over the first 90 minutes **(b)**. At the beginning, all of the organics are in the bulk vapor reservoir. The bulk vapor fraction decreases as vapors condense or are lost to the walls and claim a larger fraction of total organic mass.





the delay occurs throughout the experiment and emphasizing the importance of having a dynamic model. In Figure 6b we show the fractional product distribution for this same experiment over the first 90 minutes of the experiment by normalizing each product reservoir by the total concentration of condensible products $C^{\text{tot}} = C^v + C^t + C^d + C^s$. This confirms that the dynamical effect is greatest early in the experiment but also that a combination of steady-state condensation delay and vapor

wall losses contribute at all times. For this simulation the suspended condensation sink was a few minutes, and so the dynamical effect of the un-condensed vapors almost completely vanishes after 20 minutes, consistent with the expected equilibration time scale (Saleh et al., 2013).

Another potential explanation for the difference between CLOUD (E)LVOC yields and smog-chamber experiments is the different experimental conditions in the chambers. Specifically, the CMU experiments have reaction rates almost 3 orders of

magnitude higher than the CLOUD experiments (19 pptv s$^{-1}$ vs 0.03 pptv s$^{-1}$). As reaction rates increase, the higher frequency of collisions of intermediate products with each other may terminate the auto-oxidation reactions that create the HOMs, producing higher volatility yields than those seen at CLOUD. This may be especially important for termination reactions between peroxy radicals (RO$_2$), which are second order and will increase in importance for higher overall reaction rates. While we can not rule this out as a cause of the apparent discrepancy, we do not yet have sufficient data for the smog-chamber experiments to

test whether the apparent yield of HOMs is lower under the high-concentration conditions of the SOA formation experiments than under the more atmospherically representative experiments conducted at CLOUD. Furthermore, the original Ehn et al. (2014) plant-chamber experiments were carried out under conditions much closer to traditional smog-chamber experiments and still revealed high (E)LVOC mass yields. Conducting SOA formation experiments at very low oxidation rates is not an obvious solution, as the resulting mismatch between ambient and chamber SOA concentrations and also the very small growth rates

compared to the relatively large particle wall loss rates would make data interpretation extremely difficult; the experiments have been carried out rapidly at ambient SOA concentrations for a reason.

We can also compare CLOUD and the CMU chamber via the presence or absence of nucleation. Specifically, the CLOUD experiment was designed to observe nucleation from $\alpha$-pinene ozonolysis (Kirkby et al., 2016), whereas no nucleation occurred in the experiments reported by Pathak et al. (2007a). However, the oxidation rate in Pathak et al. (2007a) was more than two

orders of magnitude higher than that in Kirkby et al. (2016) yet the condensation sink was only about one order of magnitude higher, so the concentrations and thus saturation ratios of ELVOCs in the CMU chamber should have been higher than in CLOUD if the product yields were identical. Indeed, in Figure 7a and 7b we show the vapor supersaturation ratios from each volatility bin for Experiment 1 and Experiment 2, respectively based on the (E)LVOC yields from Kirkby et al. (2016). The ELVOC saturation ratio reaches $10^7$ for the $C^* = 10^{-8}$ $\mu$g m$^{-3}$ ELVOCs; this is an order of magnitude higher than the

saturation ratio in CLOUD (see ED Figure 7 in Tröstl et al. (2016)), confirming that nucleation should have occurred if the chemistry was identical in the two experiments.

To estimate the nucleation rates expected in the CMU experiments, we assume that new-particle formation is driven only by compounds in the $C^* = 10^{-8}$ $\mu$g m$^{-3}$ ELVOC bin, which comprises $\sim 10\%$ of the total detected HOMs in the CLOUD experiments, almost exclusively as covalently bound C$_{20}$ dimers (see ED Figure 5 in Tröstl et al. (2016)). Figure 3 in Kirkby

et al. (2016) relates the nucleation rate to the detected HOM concentration. The data in the log-log plot have a slope of 2,





indicating that the nucleation rate is a second order reaction with respect to the ELVOC concentration. Assuming that only the $C^* = 10^{-8}\ \mu\text{g m}^{-3}$ ELVOCs actually drive nucleation, we adjust the Kirkby et al. (2016) HOM concentrations down by a factor of ten and so derive a nucleation rate constant of $k_{\text{nuc}} \simeq 4 \times 10^{-14}\ \text{cm}^3\ \text{molec}^{-1}\ \text{s}^{-1}$. It is interesting to note that this is still only a small fraction of the collisional rate constant. Using the nucleation rate constant, we calculate the nucleation rate for

5  each experiment from Pathak et al. (2007a), which we show in Figures 7c and 7d. By integrating the nucleation rate over time, we find that the concentration of nucleated particles that would have formed is on the order of $10^5$ to $10^6\ \text{cm}^{-3}$. These particles would have had growth rates of hundreds of nm per hour, indicating fast growth into larger sizes that are easily detected in the SMPS. However, this was not observed. Consequently, we conclude that the experimental conditions employed by Pathak et al. (2007a) suppressed ELVOC (covalent dimer) formation relative to the conditions described by Kirkby et al. (2016). This

10  is consistent with the hypothesis that the production of ELVOCs is interrupted under higher $\alpha$-pinene concentrations, possibly through the termination of $RO_2$ auto-oxidation reactions.

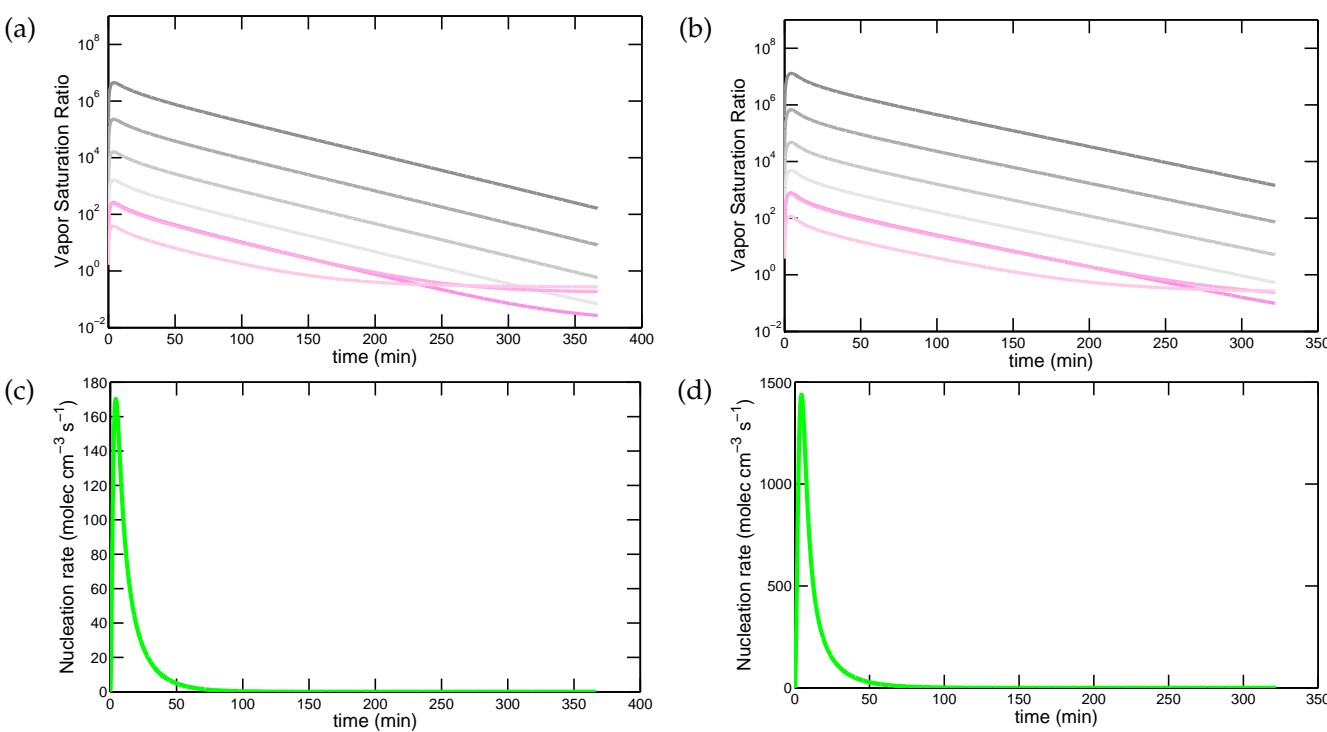

**Figure 7. Vapor saturation ratios for smog-chamber experiments experiments.** Simulated vapor saturation ratios in the CMU smog chamber for Experiment 1 (**a**) and Experiment 2 (**b**). The color of the line indicates the volatility bin, with the ELVOCs in shades of grey and LVOCs in shades of pink, and darker shades indicate lower volatility within the category. The saturation ratio of the least volatile ELVOC ($C^* = 10^{-8}\ \mu\text{g m}^{-3}$, dark gray) can be used to predict nucleation rates based on CLOUD data, which are shown in the lower panels for Experiment 1 (**c**) and Experiment 2 (**d**). However, no nucleation was observed during these experiments.



## 3.2 Oligomerization

So far our analysis has followed the base-case model of Tröstl et al. (2016), which assumes that (E)LVOC condensation drives the CLOUD growth rates and so that the nitrate CIMS sensitivity to LVOCs was low (and thus their concentrations were high). As Tröstl et al. (2016) pointed out, an alternate explanation to their growth-rate observations is that SVOC condensation

followed by oligomer formation could play a role. Oligomerization has been shown to be important to SOA formation (Kalberer et al., 2004; Tolocka et al., 2004; Heaton et al., 2009) and consistent with SOA chamber mass-yield observations (Trump and Donahue, 2014). Semi-volatile organics in the condensed phase may interact with particle phase HOMs, creating an ELVOC product. This sequesters SVOC compounds that would otherwise easily evaporate off of a particle. Furthermore, because the growth rates observed in CLOUD are small and the time constants are long (many hours), it is possible that this slow chemistry

might not be evident on the shorter timescales of the SOA formation chamber experiments we are modeling here.

As in our previous simulations we start by creating a model that matches the growth-rate results from CLOUD. There is little information on the actual yield of semi-volatile organics; thus we are merely looking to show that there is a reasonable hypothetical yield that can reproduce the CLOUD data. In this model, we start with the unscaled yields from CLOUD (the dark green in Figure 2) and add in an SVOC mass yield of 0.20 in the $C^* = 10^1$ $\mu$g m$^{-3}$ volatility bin. As a simple proof of concept

we assume that this compound will react with *any* condensed-phase organic species to form an ELVOC product. As described by Trump and Donahue (2014), the rate of dimerization is given by

$$R_{\mathrm{dimer}} = C_{\mathrm{OA}}(k_f\,w_m\,w_{\mathrm{org}} - k_r\,w_d), \qquad (8)$$

where $C_{\mathrm{OA}}$ is the organic aerosol concentration, $k_f$ is the forward rate constant of dimerization, $w_m$ is the mass fraction of monomers, $w_{\mathrm{org}}$ is the mass fraction of other organics in the particle phase (we assume the monomer reacts with all organics,

so $w_{\mathrm{org}} = 1$), $k_r$ is the dissociation rate constant, and $w_d$ is the mass fraction of dimers. For the purpose of this simple model, we assume that there is no dimer dissociation ($k_r = 0$). The CLOUD chamber operated at low $\alpha$-pinene concentrations. Thus, when we use the original, lower yield distribution, the dimerization rate must be high in order to produce the detected growth rate; we find that $k_f = 20000$ min$^{-1}$ reproduces the observations reasonably well.

Figure 8 shows the model results of the CLOUD experiments. Figure 8a and 8b shows the results of the constant HOM

experiment, and Figure 8c and 8d shows the increasing HOM experiment. In both cases, the oligomerization model reproduces the particle size and growth rate over the course of the experiments. Figure 8b and 8d shows the contributions from each of the volatility bins to the growth rate. The colors indicate the volatility of the compound, as shown in Figure 2, with the dimers shown as dark blue following Trump and Donahue (2014). The ELVOC and LVOC compounds contribute very little to the overall particle growth after the very early stages of growth, because for this simulation we assume that the nominal CIMS

concentrations are accurate. Consequently, oligomerization of SVOCs must explain the (large) residual growth. Because of the high condensed-phase rate constant, nearly all of the SVOCs that condense are immediately converted to ELVOC dimers except for the very smallest particles; simulations including a slower forward reaction simply required much higher monomer yields, which we rejected as unrealistic. The SVOC monomer does evaporate from the smallest particles because of the Kelvin enhancement. If we model the oligomerization as effectively instantaneous the growth rate for the smallest particles rises





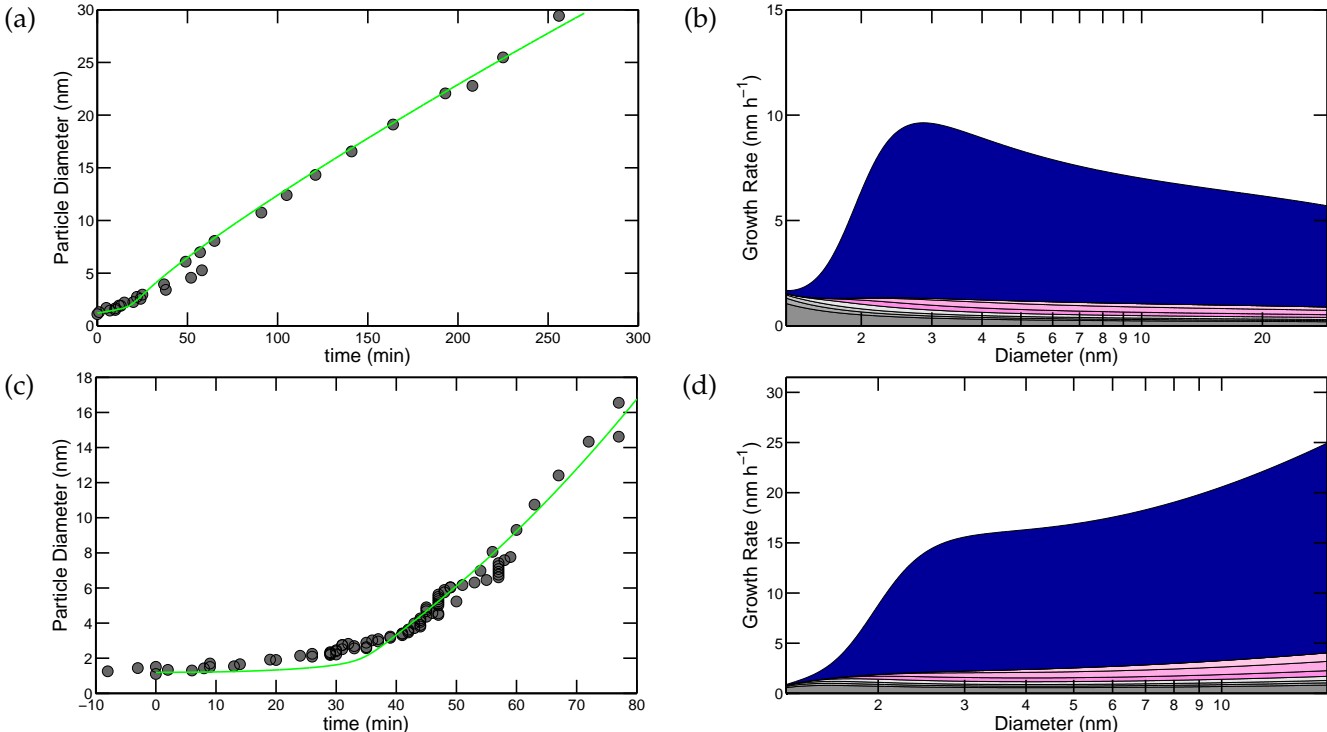

**Figure 8. Oligomerization model results for CLOUD experiments.** Oligomerizeration model particle diameters for the constant HOM **(a)** and rising HOM **(c)** CLOUD experiments, along with VBS bin contributions to growth rate vs diameter for the constant HOM **(b)** and rising HOM **(d)** CLOUD experiments. The dark blue is the contribution to growth from dimers. The acceleration near 2 nm is caused because SVOC monomers evaporate from smaller particles due to the Kelvin effect but react to form ELVOCs in larger particles.

sharply. Thus the ratio of the monomer volatility to the forward rate constant is meaningful, along with the Kelvin diameter of the SVOCs (Tröstl et al., 2016), but the individual values are practically unconstrained. While the growth-rate plot differs somewhat from the model constrained entirely by (E)LVOCs (Tröstl et al., 2016), the experimentally determined growth rate at 10 nm matches the model. Therefore, for the purposes of this exercise, this is a second product model consistent with the

5  CLOUD observations.

We can now take this oligomerization model and apply it to the SOA formation experiments from the CMU chamber. In Figure 9 we compare the base case (high LVOC) simulation with the oligomerization simulation, with the oligomerization case represented in dark green and the (extra mass from) the (E)LVOC simulation shown in light green. The oligomerization model results in a better fit to the data, though for Experiment 1 the model continues to over-predict the observations. In Figure 10 we

10  show an Odum plot for Experiment 2, reproducing Figure 5b but now also including this oligomerization simulation. There is relatively little difference between the base-case high LVOC simulation and the oligomerization simulation at low $C_{OA}$, early in the experiment, and here the model continues to overshoot the observations. This is also apparent in Figure 9b. However,



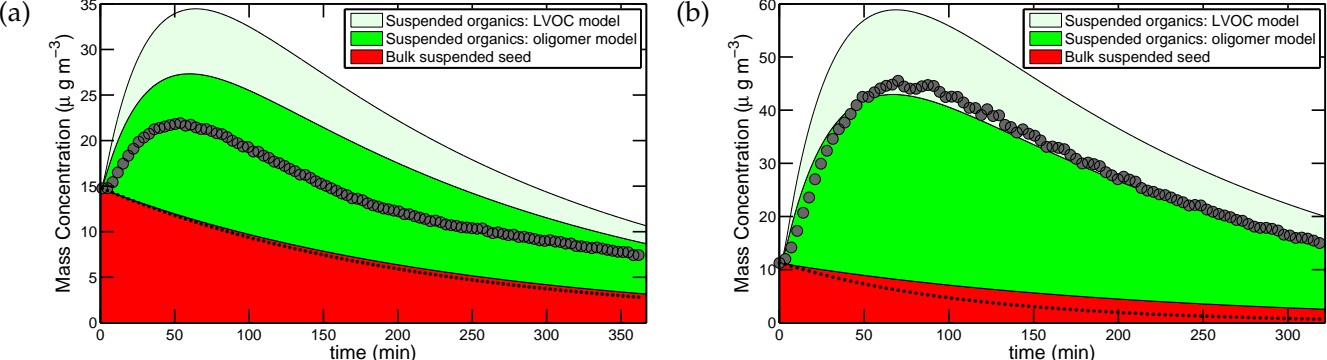

**Figure 9. Comparison of organic mass produced between an oligomerization model and the CMU chamber experiments.** Model results and data for organic (green) and seed mass (red) over time for Experiment 1 **(a)** and Experiment 2 **(b)**. Solid green is the oligomerization model while the light green shows the (additional) mass from the base-case (high LVOC) model. The oligomer model overpredicts Experiment 1, though not as much as the base-case model, and reasonably matches Experiment 2.

for the bulk of the simulation with $C_{OA} > 1\mu g\ m^{-3}$, the oligomerization simulation falls considerably closer to the data, and within the range of literature data.

In the oligomerization simulation, to reproduce the particle growth rates without excessive monomer concentrations we had to assume nearly irreversible condensation of monomers and rapid oligomerization. The volatility of the SVOC monomer is

only sufficient for evaporation to exceed reactive uptake at very small particle sizes ($d_p \lesssim 2.5$ nm) where the Kelvin effect enhances the saturation concentration sufficiently for evaporation to slow the growth. So far we have not found conditions with reversible oligomerization or slower oligomer formation that can reproduce the CLOUD growth-rate observations, though the phase space is vast. On its face, the rapid oligomerization case is not qualitatively very different from effectively non-volatile condensation, though it does reproduce the slow growth rate at very low particle diameters observed in the CLOUD experiment.

It is thus somewhat surprising that the mass yields in the oligomerization simulations are significantly lower than the (E)LVOC case. Most of this difference is because a flux balance differs from a mass balance. The SVOC monomers are relatively light, with $M_i = 175$ g mole$^{-1}$ as compared to (E)LVOCs with $M_i \simeq 350$ g mole$^{-1}$. This means that for the same vapor mass concentration, the SVOC monomers have a 44% higher condensation rate, simply because they have a higher velocity.

### 3.3 The condensation sink and reaction rates

At this point we have a dynamical model that can reproduce the growth-rate observations from CLOUD while not grossly over-predicting the SOA mass production rate observed in at least some SOA formation chamber experiments. However, the model still leaves no room for true SVOC condensation (save for nearly irreversible conversion to oligomers), and so it is not yet fully consistent with observations strongly suggesting that 30-60% of the SOA in chambers is semi volatile. We thus can not rule out possible changes to the gas-phase chemistry (and the volatility distribution of the products); this is difficult without

corresponding measurements of gas-phase HOMs via nitrate-CIMS in the chamber experiments. Indeed, our simulations of



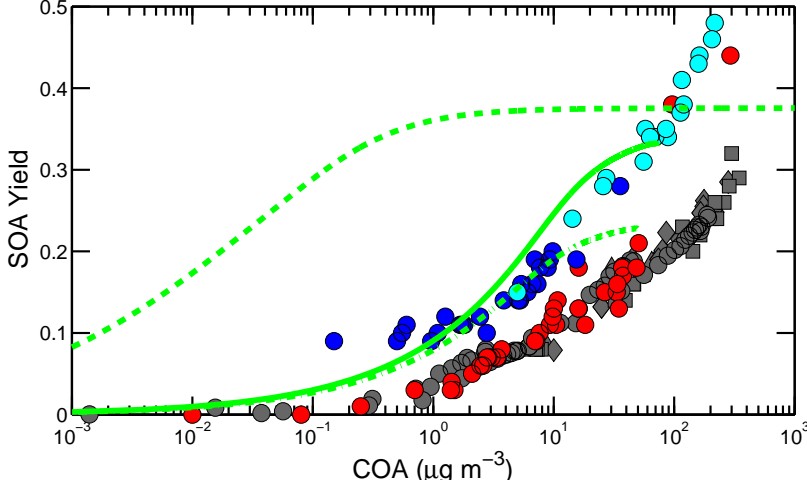

**Figure 10. Odum plot with the LVOC and oligomer models for Experiment 2.** The LVOC model is a solid green curve and the oligomer model is a dot-dashed green curve. The dashed green curve is the equilibrium partitioning yield. The data from Pathak et al. (2007a) (red circles) are directly comparable. Other literature data are from Presto and Donahue (2006) (grey), Shilling et al. (2008) (blue triangles), and Song et al. (2007) (cyan). The oligomer model is consistent with some prior experiments.

nucleation in the CMU experiments, and the absence of nucleation in the data, strongly suggest that in the very least the ELVOC yields were lower in the CMU experiments than in the CLOUD experiments.

Other dynamical effects also remain possible explanations. One possibility is that the mass accommodation coefficients differ in the high-mass SOA formation experiments and in the low-mass CLOUD experiments. However the flux-balance constraints for CLOUD strongly suggest a mass accommodation coefficient near unity. Specifically, the total mass yields required to explain the growth rates already stretch plausibility, and $\alpha < 1$ would only require higher vapor concentrations (and thus higher yields) to compensate for the lower specific condensation rate. However, if larger particles had a lower effective mass accommodation coefficient (for example driven by slow particle-phase diffusion), that might direct more vapors to the walls and lower the overall observed SOA production. We explore this by varying the particle condensation sink in our simulations, using the high (E)LVOC simulations as our base case.

The particle condensation sink is key to condensing organic vapors, and in chamber studies, this condensation to suspended particles is in competition with the loss of vapors to chamber walls. The key to capturing oxidation products is therefore increasing the condensation sink by having a higher seed surface area. In general our design objective is to have a suspended seed condensation sink at least $10\times$ greater than the vapor-wall collision frequency. For the CMU chambers, with a vapor wall-loss frequency of approximately $0.1\ \mathrm{min}^{-1}$ (Ye et al., 2016a), this means that the ideal seed condensation sink is of order $1\ \mathrm{min}^{-1}$. The critical parameter is the ratio of the seed condensation sink to the wall-loss rate constant.

Figure 11 shows the locations of vapors given an initial particle-to-wall condensation sink (CS) ratio for a hypothetical mix of 3 ppb $\alpha$-pinene and 50 ppb ozone. The CS ratio describes how likely an organic vapor molecule is to hit (and condense





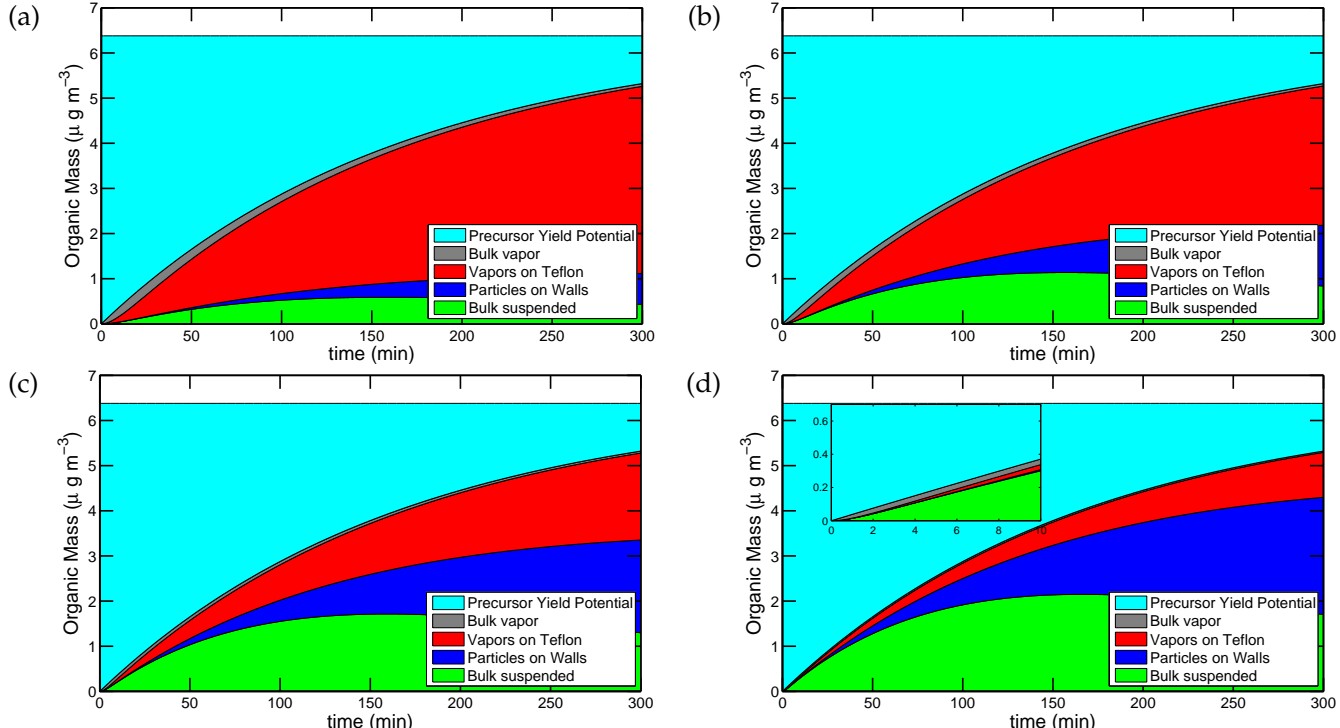

**Figure 11. The effect of varying particle-to-wall condensation sink ratio by varying the number of seeds at 3 ppb $\alpha$-pinene and 50 ppb ozone.** The CS ratio is the ratio of the suspended condensation sink to the vapor wall-loss rate constant. CS ratio = 0.3 (**a**); CS ratio = 1 (**b**); CS ratio = 3 (**c**); CS ratio = 10 (**d**). The key area is the grey area denoting the bulk vapor (the sliver between the teal and red), which comes from reacted products that have not yet condensed into the particle phase. For a CS ratio of 10, there is sufficient seed concentration to condense most of the vapors into the particle phase (mass of particles on walls and bulk suspended). Conversely, for a CS ratio of 0.3, the lack of seeds causes a buildup of bulk vapor, most of which is then lost to the walls.

to) a particle versus the wall, and in broad terms for this simulation where the condensible vapors are effectively non volatile, the CS ratio also gives the ratio of the condensation flux to the particles (the sum of the green and blue in Figure 11) and the wall loss (the red in Figure 11). For low CS ratios the condensible vapors build up (the gray in Figure 11a), showing a delay of condensation of vapors to particles. The majority of these vapors are thus lost to the walls. Even in Figure 11b, where the

5   initial CS ratio is 1:1, more of the mass is lost to the walls than is condensed onto particles, though the ratio is close to 1:1. The slightly higher vapor wall loss is due to particle wall loss, which decreases the available surface area in the bulk chamber. As CS ratio increases, the bulk vapor concentrations decrease as higher particle condensation rates collect most of the organic mass.

The CS ratio is dependent on the total suspended surface area and thus both the particle number concentration and the particle
10   diameters. Over the course of an experiment, there are competing processes that affect the CS through these two variables. The particle number concentration decreases due to particle wall loss. The particle diameter increases due to condensation.





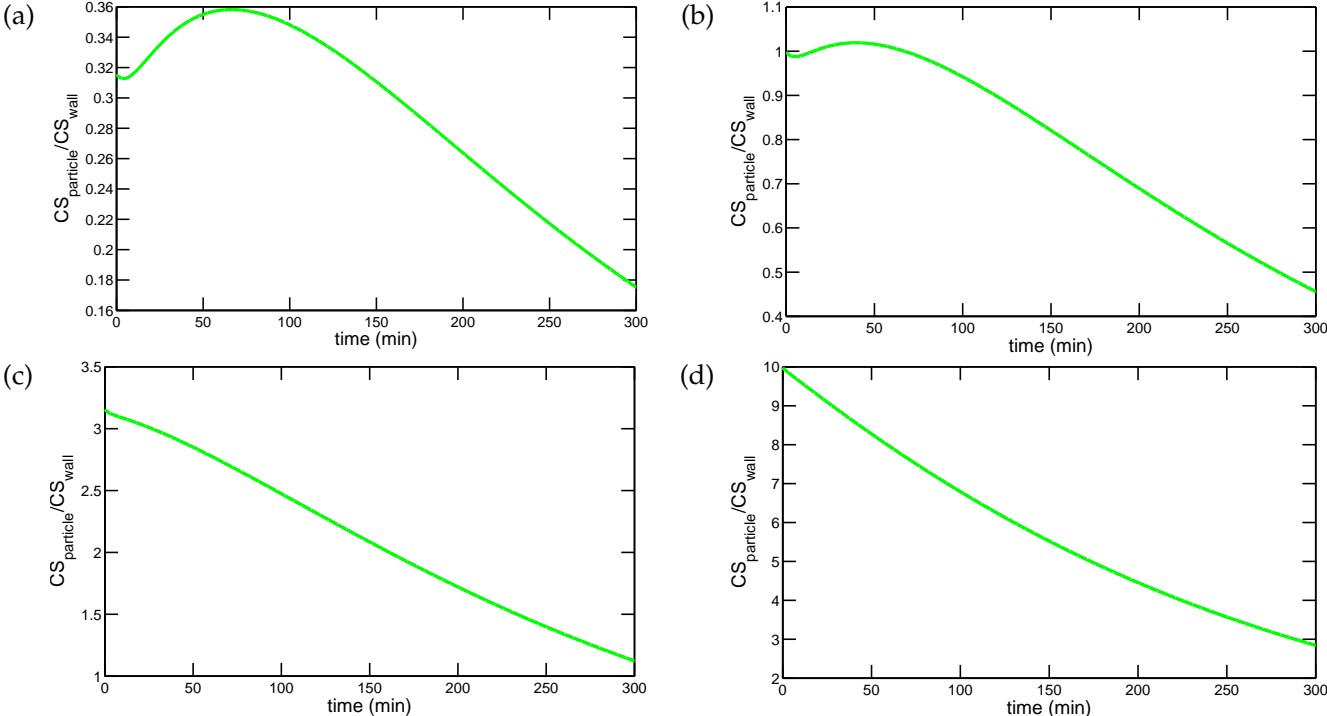

**Figure 12. Evolving condensation sink (CS) ratio over the course of a chamber run, depending on the initial CS ratio.** Each run has the same initial condition: 3 ppb $\alpha$-pinene and 50 ppb $O_3$. The number of seeds is increased by one-half a decade each from case to case by increasing the particle number concentration. CS ratio = 0.3 (**a**); CS ratio = 1 (**b**); CS ratio = 3 (**c**); CS ratio = 10 (**d**). At a low CS ratio, fewer seeds mean that each seed grows substantially from condensation. Because the CS depends on the surface area of each particle, a faster growth of the surface area results in an increase in the particle condensation sink. At high seed concentrations, the initial rise in the CS ratio does not occur because each seed receives little organic mass. However, CS is also dependent on the particle number concentration. Therefore, the CS ratio steadily drops over the course of a chamber run as particles themselves are lost to the walls. The specific value of CS ratio at which the growth rate effect disappears is dependent on the amount of precursor and the oxidation rate of precursors.

However, the rate at which the diameter increases is also dependent on the particle number. In Figure 12 we show the evolution of the CS ratio over time for the four CS-ratio simulations. At low CS ratios (Figure 12a), or low particle concentrations, condensation has a greater effect on the diameter of each particle. This causes the CS ratio to increase initially before decreasing later from particle wall loss. For simulations with a higher initial CS ratio, the effect of particle diameter is lessened as the growth rate of each individual particle is slower. By the simulation shown in Figure 12d, the diameter growth effect is negligible.

In addition to the CS ratio, the condensation rate is affected by the reaction rate. The reaction rate is simply the product of the reaction rate constant, the $\alpha$-pinene concentration, and the ozone concentration. With more rapid oxidation more condensation occurs early, before substantial particle wall losses deplete the condensation sink. Furthermore, the steady state saturation ratios will be correspondingly higher. Figure 13a and Figure 13b show the reservoirs of organic products at 500 ppb ozone and





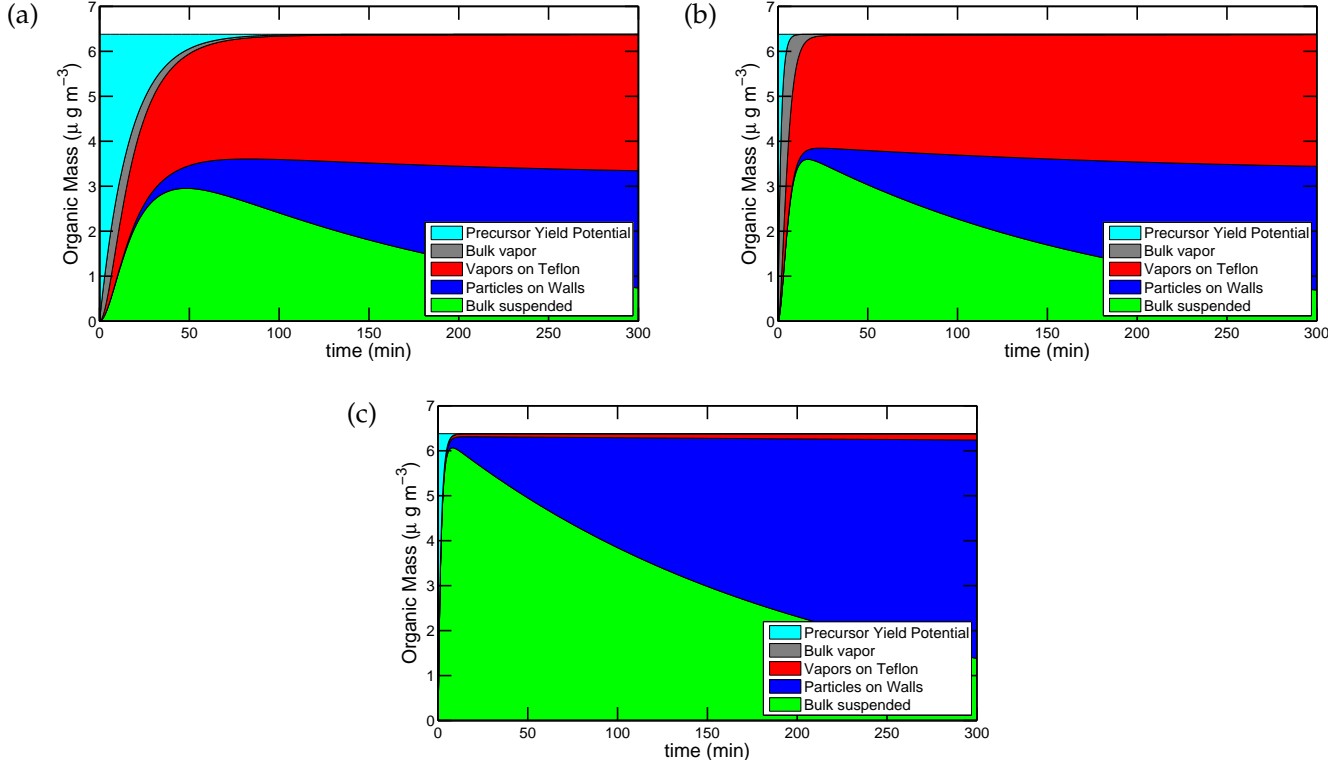

**Figure 13. Locations of organic products from 3 ppb $\alpha$-pinene with varying concentrations of ozone and CS ratios.** Modest ozone, modest CS ratio (500 ppb $O_3$ with a CS ratio of 1, **a**); High ozone, modest CS ratio (5000 ppb $O_3$, CS ratio of 1, **b**); High ozone, high CS ratio (5000 ppb $O_3$, CS ratio of 100 **c**). Given a seed concentration, increasing the ozone concentration causes $\alpha$-pinene to react faster, resulting in a higher condensation driving force and a higher organic particle mass after a shorter period. However, half of the vapors are still lost to the walls. By increasing the CS ratio, or seed concentration, by two orders of magnitude, all of the condensible vapors can be captured in the particle phase.

5000 ppb ozone, respectively. Compared to Figure 11b, which has the same $\alpha$-pinene and seed concentrations, here organic aerosol mass is formed faster and concentrations are higher. Even though the LVOCs have very low volatility, we can also see evaporation of organics off the particles toward the walls as the run continues, as $C^t$ grows while $C^s + C^d$ shrinks. However, we would like to avoid vapor wall losses altogether, if possible. Figure 13c shows that in theory it is possible to minimize the

5   wall loss by increasing both the reaction rate and the condensation sink in the chamber. In this case, almost all of the organics condense to particles before slowly being lost to the walls. It is trivial to extrapolate the green condensed-phase concentration back to the "correct" value; unfortunately, this comes at the expense of running the chemistry extremely quickly, and potentially perturbing the gas-phase chemistry (especially the yields of HOMs due to auto-oxidation) and also almost certainly driving intense nucleation.





# 4   Conclusion

In this work, we took a dynamical 1-D volatility basis set model developed to model growth rates of freshly nucleated particles measured in CLOUD experiments at CERN and adapted it to $\alpha$-pinene ozonolysis experiments addressing SOA mass formation conducted in the CMU smog chamber. Based on the mass yield distribution from CLOUD, we found that our model

overpredicts the organic mass produced and the resulting SOA mass yields over the course of these typical SOA formation experiments. However, we demonstrated that chamber experiments need to be treated dynamically, because there is a delay between the formation of low volatility vapors and the condensation of these vapors to particles. This delay at least partially resolves the issue of the existence of low volatility compounds that do not seem to show up in Odum plots – they exist, but show up at higher aerosol loading than expected because of the time it takes for them to condense.

We found that substantial oligomerization is consistent with both the CLOUD and the CMU chamber results. By allowing semivolatile organics in the condensed phase to form dimers with lower volatility compounds, we showed that it is possible to replicate the data from CLOUD experiments. It is likely that oligomerization plays a role in organic aerosol formation, but how substantial a role remains to be determined. Because of the many parameters available to explain the current set of observations (HOM yields, oligomer fraction, mass accommodation coefficients, vapor wall losses, $RO_2$ auto-oxidation rates, etc.), only a

very carefully designed series of experiments will fully constrain this problem. Whether high LVOCs or oligomerization is responsible for the CLOUD growth-rates, we would expect to have observed nucleation in the CMU chamber experiments, where none occurred; this strongly suggests that the gas-phase product distributions in the two experiments are different, though the reasons remain uncertain.

We emphasize that the ratio of vapor-particle condensation sink to the vapor-wall loss sink is critical to interpretation of

smog-chamber data. At low initial CS ratios, most of the organic vapors produced are lost to chamber walls. As the CS ratio increases, more of the mass goes to the particles, but the suspended mass concentration does not scale with the CS ratio. Because of particle wall loss, the organics on suspended particles are driven to the walls. For the same reason, the condensation sink to the remaining particles also decreases over time. Therefore, merely increasing the condensation sink does not always increase the concentration of bulk organic particle mass. Ideal chamber conditions require both high CS ratios and

high oxidation rates (by boosting ozone concentrations). At high oxidation rates, all of the $\alpha$-pinene is immediately reacted into low volatility compounds, and the high CS ratio allows these compounds to quickly condense onto seed particles. This allows all the organics to be collected onto seeds before wall losses in either the vapor or suspended phase can have a large effect. However, this condition may in turn interfere with the unimolecular gas-phase auto-oxidation chemistry that produces the HOMs in the first place. Consequently, direct measurements of the gas-phase HOM yields during such experiments are

critical to the overall interpretation of the experimental data.

*Acknowledgements.* This work was sponsored by the U.S. National Science Foundation grant AGS 1447056.





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
