# Peer review of "Dynamic Consideration of Smog Chamber Experiments"

_Atmospheric Chemistry and Physics, 2016_

## Referee Comment (RC1) · Anonymous Referee #1 · 19 Dec 2016

General comments

This manuscript explores an important and previously unexplained paradox in secondary organic aerosol (SOA) formation, namely that chamber experiments suggest that the majority of SOA is semi-volatile while recent studies focused on particle nucleation show high mass yields of very low volatility compounds. The authors present a comprehensive modeling analysis of previously published experimental data in an attempt to explain this paradox. They propose and test several possible theories to explain the paradox and conclude by suggesting further experiments needed to fully resolve it. The manuscript fits very well within the scope of ACP and is scientifically sound. I think the manuscript is suitable for publication in ACP, but further clarification/explanation is needed in several parts of the manuscript as well as minor technical corrections.

Major comments

1. Page 6, lines 1-11: Is the rate of vapor wall loss assumed to be the same for both the CMU and CLOUD chamber? Is this a reasonable assumption? What is the size difference between the two chambers? If the surface-to-volume ratio between the two chambers is significantly different, the vapor wall loss rates may also differ substantially. How might a significantly different vapor wall loss rate between the two chambers affect the overall conclusions of the manuscript?

2. Page 6, lines 13-24: I found this paragraph describing particle wall loss vague and somewhat confusing. Please explain more clearly what $k^{s,d}$ is and how it is defined. On line 19, the authors state "we rely on the seed loss rate measured prior to the injection of α-pinene." Is this the total number or total volume loss rate? Is $k^{s,d}$ defined as this loss rate? One line 20, what is meant by "minor discrepancies between the data and the model concerning the mass of seeds in the chamber"? Does the model over- or underestimate the mass of seeds in the chamber? Why does this discrepancy exist since the mass of seeds is not directly measured by the SMPS? If the measured loss rate is used to extrapolate the initial mass of seeds in the chamber and then this loss rate is applied in the model, I would assume that the modeled and 'measured' (which must still be corrected by this loss rate) mass of seeds would be identical.

4. Page 17, line 12-13: What is the significance of a higher condensation rate? Do the authors mean to suggest that there is a higher nucleation rate for a lower mass yield? Wouldn't a higher condensation rate also increase the yield for the CMU experiment?

5. Page 17, section 3.3: How does this section affect the conclusions regarding discrepancies in volatility distributions between the CMU and CLOUD experiments? At the beginning of this section, page 18, line 7-10, the authors state, "if larger particles had a lower effective mass accommodation coefficient (for example driven by slow particle-phase diffusion), that might direct more vapors to the walls and lower the overall observed SOA production. We explore this by varying the particle condensation sink in our simulations." However, the results of these simulations are never connected back to this initial hypothesis. Do these simulations support the idea that the lower yield in the CMU chamber could be due to changes in the particle condensation sink? In its current form, this section does not contribute to the overall attempt to reconcile the CLOUD and CMU experiments.

Minor comments

1. Throughout the manuscript, the authors switch between using the words "dynamic" and "dynamical." These words are similar but in some instances have slightly different meanings. I would recommend choosing one version and using it throughout the paper.

2. Page 1, line 10: What experiments is the phrase "substantially lower SOA mass yields" referring to? The CMU SOA experiments or the CLOUD experiments? If referring to the CLOUD experiments, I recommend mentioning in the abstract how the second oligomerization model behaves for the CMU experiments.

3. Page 1, lines 22-25: This sentence is very long and somewhat confusing. It would be easier for the reader if it was broken up into two sentences.

4. Page 3, Figure 1: What are the small black arrows drawn between the volatility bins? They are not mentioned in the text or the caption and resemble a reverse aging scheme. I recommend explaining what is meant by the arrows or removing them.

5. Page 5: The organization between the "Dynamic Model" and "Results and Discussion" section could be clearer. Specifically, I would prefer to see a subsection within the "Dynamic Model" section (which could possibly be renamed "Methods") describing the experimental conditions of both the CLOUD and CMU experiments (i.e. moving the first paragraph from "Results and Discussion" here and adding a brief description of the CLOUD experiments). It will be helpful to the reader to have both experiments described in the same section for later reference.

6. Page 5, line 21: I believe "low volatility" is meant in in place of "semi-volatile": the volatility bin that is referenced is $10^{-1}$ µg m$^{-3}$ and the corresponding abbreviation in parenthesis is "LVOC".

7. Page 6, section 2.2: It was unclear if the volatility distribution used was taken directly from Tröstl et al. (2016) or if the authors performed the fit themselves. Furthermore, was the temperature correction done by Tröstl et al. (2016) or the authors?

8. Page 6: lines 31-32: I found the phrasing of this sentence awkward and had to reread it several times to understand it.

9. Page 8: equation (5): What are $v_{i,p}$ and $B_{i,p}$? These are not defined in the subsequent paragraph. What value is used for the accommodation coefficient $\alpha_{i,p}$?

10. Page 9, line 4: I believe "(E)LVOC **vapor** wall loss" is meant in place of "(E)LVOC **particle** wall loss."

11. Page 9, Figure 3: It is unclear how the light blue area, the Precursor Yield Potential, is calculated. The caption states that it is unreacted precursor, but line 7 of the following page states it is "the concentration of organics that have not yet been formed by ozonolysis." How is this calculated?

12. Page 11, line 9: What is a "perfect correction for the deposited particle mass"? Is this related to the lower or upper limit assumption (i.e., Loza et al., 2012)?

13. Page 15, lines 24-15: What are the "constant HOM experiment" and the "increasing HOM experiment"? These were not described previously.

14. Page 18, line 10: Please describe more explicitly what the base case is. Does the high (E)LVOC simulation include the additional LVOC concentration required to explain the observed growth rates in CLOUD?

Loza, C. L.; Chhabra, P. S.; Yee, L. D.; Craven, J. S.; Flagan, R. C.; Seinfeld, J. H. Chemical aging of m-xylene secondary organic aerosol: laboratory chamber study. Atmos. Chem. Phys. 2012, 12, 151−167.

---

## Referee Comment (RC2) · Anonymous Referee #2 · 5 Jan 2017

Review of "Dynamic Consideration of Smog Chamber Experiments" by Wayne K. Chuang and Neil M. Donahue

The study aims to provide insights to the apparent contradiction in observations of secondary organic aerosol (SOA) formation in smog-chamber and nucleation-chamber experiments. The authors apply a dynamic volatility basis set (VBS) model to explore possible reasons for why smog-chamber experiments suggest that SOA from $\alpha$-pinene is mainly semi-volatile, while recent nucleation-chamber experiments at CLOUD point to high yields of very low-volatile organics.
The study is well conducted and suits in the scope of ACP. The used methods and obtained results are clearly presented, and the conclusions are generally reasonable. There are, however, some additional aspects relevant for the interpretation of the

experimental data that are not brought up in the study. In addition, it is not clear if the model used in the study can provide a valid description of particle growth at all different conditions and throughout the studied particle size range. Publication in ACP can be considered after the authors have addressed the issues listed below. I also agree with the main comments of Referee 1.

Specific comments:

Major comments:

1) The manuscript is motivated by the fact that yields of extremely low-volatile products in the CLOUD experiment are apparently much higher than in previous smog-chamber experiments. How large are the uncertainties in the assessed volatilities? Are the saturation vapor pressures of the $\alpha$-pinene oxidation products based on their O:C ratios and the SIMPOL model, as presented by Tröstl et al. (2016)? The recent study "$\alpha$-pinene autoxidation products may not have extremely low saturation vapor pressures despite high O:C ratios" by Kurtén et al. (2016) shows that group-contribution methods such as SIMPOL very likely underestimate the volatility of $\alpha$-pinene autoxidation products. SIMPOL predictions for the saturation vapor pressures of species studied in the work of Kurtén et al. are generally lower compared to other methods that are more suitable for describing the chemical interactions of autoxidation systems.
How would the comparison of the CLOUD and smog-chamber experiments look like if the saturation vapor pressures of the species of the CLOUD chamber were not as low as assumed by Tröstl et al.?

2) When modeling the particle size evolution at the conditions of the CLOUD experiment, the role of ions and ion-ion recombination in the growth of the particle population is not addressed at all. Kirkby et al. (2016) state that the new-particle formation

(NPF) events generated in the CLOUD chamber, from which the experimental growth rate (GR) data used in the present manuscript (as well as in the study by Tröstl et al. (2016)) are extracted, are in practice driven by ions when the ion clearing field is off. The nucleation rates presented by Kirkby et al. imply that almost all particles that appear at $d_p$ = 1.7 nm at a HOM concentration of around $10^7$ cm$^{-3}$ are either positively or negatively charged, and "ion–ion recombination progressively neutralizes the charged particles as they grow".

I assume that at least the "constant HOM" data is from the CLOUD ion runs, as I don't see how the HOM concentration could otherwise be kept constant in the experiment while observing a NPF event. If this is so, the experimental growth rates determined from the time evolution of size-classified particle concentrations should be influenced by

-Recombination of particles of opposite polarities to form larger, electrically neutral particles.

-Possible enhancement in the condensation rate of molecules onto particles due to electrostatic interactions (charge enhancements for different compounds in the CLOUD chamber have been assessed by Lehtipalo et al. (2016)).

3) Even when ions are not involved, coagulation of molecular clusters and small particles may also play a role in the evolution of the particle distribution. In the CLOUD experiment, this was previously addressed for NPF from sulfuric acid and bases by Lehtipalo et al. (2016), who concluded that nano-particle growth can be assisted by cluster coagulation in the presence of a strong stabilizing compound. While Lehtipalo et al. studied different chemical compounds than used here, the conclusion on the role of coagulation is general: the less small particles evaporate, the more their concentrations increase and thus the more significant the effect of coagulation becomes. This should be relevant also for NPF from oxidized organics, if ELVOCs are capable of forming very stable small clusters -can the authors assess the effect of coagulation at the studied conditions?

4) In the CLOUD experiment comparisons, the particle growth model is applied to particle sizes down to around one nanometer. However, the validity of a single-particle model at very small nano-particle sizes is questionable. The model is "deterministic" in the way that it assumes that all particles of the same size grow at the same rate. It can't thus describe the fact that while the very smallest particle sizes may be unstable against evaporation so that the evaporation rate of molecules from the particle exceeds the collision rate of molecules, these unstable particles can still grow and generate a flux of particles towards larger sizes (this is the definition of nucleation). Instead, the used model requires the condensation rate to exceed the evaporation rate for all particle sizes in order for the particles to grow, even if this might not be the case in reality.

It appears that the authors are trying to simulate the initial NPF process with a model that might not even in principle be capable of describing the phenomenon. Is it reasonable to start modeling the growth from the size of $d_p \approx 1$ nm, which corresponds to approximately a molecule or two? Is the growth rate of a single molecule even a reasonable concept?

5) Further, the evaporation of molecules from particles is described by the Kelvin approach throughout the modeled size range, and it is concluded that the Kelvin effect is important especially for the smallest sizes. On the other hand, the Kelvin approximation is based on classical macroscopic droplet thermodynamics, and is therefore not expected to be valid for small sizes of a few molecules, which instead need to be treated with more sophisticated, atomic-level methods (e.g. Merikanto et al., 2007). Evaporation rates based on the Kelvin approach are thus likely to be highly uncertain; how does this uncertainty affect the results?

Similarly, can condensed-phase oligomerization processes be expected to be similar for larger, macroscopic particles and for the smallest modeled sizes consisting of only a few molecules? Can the smallest sizes even be considered particle phase?

6) The contribution of different VOC species to particle growth is obtained by fitting the model to experimental growth rates deduced from the "appearance times" of

different particle sizes. Recent studies suggest that for the smallest, sub-3 nm sizes, the experimental appearance time-GR method can give distorted results, namely the apparent growth rates can be too high (Olenius et al., 2014; Kontkanen et al., 2016). Would the VBS fit change if the GRs of sub-3 nm sizes were lower?

Minor comments:

1) In the introduction, the expression "yield" is used in the context of both SOA mass yields and (E)LVOC yields; for easier reading, it would be good to better clarify which quantity is addressed when discussing the yields.

2) Page 12, lines 4-5: Please explain CSTR mode and batch mode, and also reformulate the expression "significantly higher than the (older) data" which is somewhat unclear.

3) Page 13, line 5: The expression "For this simulation the suspended condensation sink was a few minutes" is not all clear; please clarify what is meant by a sink being a few minutes.

4) Page 13, line 35: When discussing the nucleation rate data from Kirkby et al., are the authors referring to the neutral or ion-induced rates?

5) Page 14, lines 1-5: "The data in the log-log plot have a slope of 2, indicating that the nucleation rate is a second order reaction with respect to the ELVOC concentration": generally, the slope does not indicate something about reactions (e.g. Ehrhart and Curtius, 2013). The derived "nucleation rate constant" doesn't have an actual physical meaning, as the rate is not affected only by the vapor concentration (and cluster evaporation), but also by e.g. losses of the formed molecular clusters onto chamber walls and surfaces of larger particles. The slope of the nucleation rate, regardless of what all the dynamic processes affecting the nucleation are, also varies with the absolute vapor

concentration (as well as other factors affecting the nucleation process), and therefore applying the "nucleation rate constant" to a completely different experimental set-up is questionable.

Large sinks, i.a. large suspended particles, can efficiently suppress nucleation; could this be related to the fact that particle formation was not observed in the smog-chamber experiments? Moreover, how was the effect of temperature on the nucleation rate taken into account when assessing rates for a smog-chamber?

6) Figure 8: Why are the growth rates of specific particle sizes much higher in the "increasing HOM" case? Are the absolute vapor-phase HOM concentrations similar for the "constant HOM" and "rising HOM" experiments? Why is the particle size range different for the two experiments? Same x- and y-axis limits would make the figure easier to read. Why are some particle appearance times negative in panel c? How is $t = 0$ defined? It is stated that "the experimentally determined growth rate at 10 nm matches the model", but no experimental growth rates are presented; please show also GRs deduced from the CLOUD observations in Figure 8.

7) Are the presented particle diameters mass or mobility diameters?

Technical comments:

-Page 11, line 10: Should "$\Delta\alpha\text{-pinene}/C_{\text{OA}}$" be "$C_{\text{OA}}/\Delta\alpha\text{-pinene}$"?

-Page 12, line 4: Remove "also" from the sentence "In Figure 5 we also also show..."

-Figure 1: Explain the markings with alphas on the right-hand side of the figure frame. Also state that the figure (minus the offset bar and the green curve) is taken from Presto and Donahue (2006).

-Figure 3, figure caption: Expressions "Experiment 1 monodisperse (a) and polydisperse (b) results. Experiment 2 monodisperse (c) and polydisperse (d) results." are

somewhat clumsy; please replace them with proper sentences.

-Figure 6: The upper limit of the y-axis could be set to 1, as the white space above the value of 1 is unnecessary.

-Figure 7 a and b: A legend would be helpful. Also, an extra occurrence of the word "experiments" should be removed from the figure caption: "Vapor saturation ratios for smog-chamber experiments experiments", and "dark gray" should be "dark grey". The unit of the nucleation rate on the y-axis label should be particles cm$^{-3}$ s$^{-1}$, not molec. cm$^{-3}$ s$^{-1}$.

-Figure 8: A legend would be good here, too. In the caption, "oligomerizeration" should be "oligomerization".

-Figure 10: In the caption, "blue triangles" should presumably be just "blue".

-Figure 12: I don't see the discussion starting with "At a low CS ratio, fewer seeds..." belonging to a figure caption (and in any case, essentially the same information is already included in the main text, so there is no need to repeat it). Similarly for Figure 13: the explanation starting with "Given a seed concentration..." shouldn't be a part of a caption.

-For all figures that present same quantities for different experimental set-ups or simulation cases in separate panels (Figures 3-5, 7-9, and 11-13), adding titles to the panels would make the figures easier to read.

-There are quite many separate figures in the manuscript. Could some of the figures be combined together as panels of a larger figure (e.g. Figures 4 and 5), or merged into the same panel (e.g. Figure 5 and 10)?

References:

Ehrhart and Curtius, Atmos. Chem. Phys. 13, 11465–11471, 2013
Kirkby et al., Nature 533, 521–526, 2016
Kontkanen et al., Atmos. Chem. Phys. 16, 5545–5560, 2016
Kurtén et al., J. Phys. Chem. A 120, 2569–2582, 2016
Lehtipalo et al., Nat. Commun. 7, 11594, 2016
Merikanto et al., Phys. Rev. Lett. 98, 145702, 2007
Olenius et al., J. Aerosol Sci. 78, 55-70, 2014
Presto and Donahue, Environ. Sci. Technol. 40, 3536–3543, 2006
Tröstl et al., Nature 533, 527–531, 2016
* * *

---

## Author Response (AR1)

Referee 1:

*We have modified the text of our manuscript in almost all cases following the suggestions of the reviewer, as described below.*

Major comments

1. Page 6, lines 1-11: Is the rate of vapor wall loss assumed to be the same for both the CMU and CLOUD chamber? Is this a reasonable assumption? What is the size difference between the two chambers? If the surface-to-volume ratio between the two chambers is significantly different, the vapor wall loss rates may also differ substantially. How might a significantly different vapor wall loss rate between the two chambers affect the overall conclusions of the manuscript?

*The vapor wall loss rates for both chambers have been measured and are documented in the literature. They are very similar (.09 min$^{-1}$ for the CLOUD chamber (Kirkby et al., 2016) and 0.1 min$^{-1}$ (Ye et al., 2016) and we have modified the text to make this clear. These values are close enough (and likely variable enough) that for the sake of simplicity in the presentation we chose to represent them as identical for our simulations; the minor differences do not significantly affect our conclusions.*

2. Page 6, lines 13-24: I found this paragraph describing particle wall loss vague and somewhat confusing. Please explain more clearly what $k^{s,d}$ is and how it is defined. On line 19, the authors state "we rely on the seed loss rate measured prior to the injection of $\alpha$-pinene." Is this the total number or total volume loss rate? Is $k^{s,d}$ defined as this loss rate? One line 20, what is meant by "minor discrepancies between the data and the model concerning the mass of seeds in the chamber"? Does the model over- or underestimate the mass of seeds in the chamber? Why does this discrepancy exist since the mass of seeds is not directly measured by the SMPS? If the measured loss rate is used to extrapolate the initial mass of seeds in the chamber and then this loss rate is applied in the model, I would assume that the modeled and "measured" (which must still be corrected by this loss rate) mass of seeds would be identical.

*The "discrepancies" described here are an inevitable product of the monodisperse simulations; in the monodisperse case, we constrain the initial seed mass and surface area (condensation sink) based on the "condensation sink diameter" as described in the text. However, as the particles grow, it is impossible to conserve both mass and surface area; this is the main reason we include the polydisperse (moving mode) simulations as well.*

*In our simulations, the rate constant $k^{s,d}$ is the rate of particle mass loss to the chamber walls, constrained by the observed loss rate of seed volume prior to the start of condensation. The discrepancy between the model and actual seed concentrations arises from the use of a monodisperse model to estimate a polydisperse seed distribution. In order to accurately model condensation, a monodisperse seed diameter is chosen to match the total condensation sink of the*

*polydisperse seeds. The seed mass derived from the monodisperse diameter is different than the actual mass, and this difference remains through the model run.*

4. Page 17, line 12-13: What is the significance of a higher condensation rate? Do the authors mean to suggest that there is a higher nucleation rate for a lower mass yield? Wouldn't a higher condensation rate also increase the yield for the CMU experiment?

*A higher condensation rate does not necessarily increase the mass yield. The high volatility of SVOC monomers implies that they also have a high evaporation rate. The higher SVOC condensation rate influences the mass yield only when there is also a high rate of oligomerization, where the SVOC becomes captured in the aerosol phase as dimers.*

5. Page 17, section 3.3: How does this section affect the conclusions regarding discrepancies in volatility distributions between the CMU and CLOUD experiments? At the beginning of this section, page 18, line 7-10, the authors state, "if larger particles had a lower effective mass accommodation coefficient (for example driven by slow particle-phase diffusion), that might direct more vapors to the walls and lower the overall observed SOA production. We explore this by varying the particle condensation sink in our simulations." However, the results of these simulations are never connected back to this initial hypothesis. Do these simulations support the idea that the lower yield in the CMU chamber could be due to changes in the particle condensation sink? In its current form, this section does not contribute to the overall attempt to reconcile the CLOUD and CMU experiments.

*This section is meant as a discussion of possible explanations that as yet are insufficiently constrained. We have modified the text to make this clearer. The mass accommodation coefficient is merely one of numerous factors that need to be considered in future experiments to reconcile the CMU and CLOUD experiments. It would be premature for us to say how much, or even whether, the mass accommodation coefficient can explain the differences between the experiments. Rather, these simulations show the effect of chamber conditions on organic vapor wall loss and what researchers should consider when designing their experiments.*

Minor comments:

1. Throughout the manuscript, the authors switch between using the words "dynamic" and "dynamical." These words are similar but in some instances have slightly different meanings. I would recommend choosing one version and using it throughout the paper.

*We now use "dynamic" exclusively.*

2. Page 1, line 10: What experiments is the phrase "substantially lower SOA mass yields" referring to? The CMU SOA experiments or the CLOUD experiments? If referring to the CLOUD experiments, I recommend mentioning in the abstract how the second oligomerization model behaves for the CMU experiments.

*In this paper, the mass yield is independent of the experiment performed, and is a property of the reaction of $\alpha$-pinene with ozone. "Substantially lower SOA mass yields" would therefore be applied to both experiments. We have added the result of the oligomerization model in the CMU chamber to the abstract.*

3. Page 1, lines 22-25: This sentence is very long and somewhat confusing. It would be easier for the reader if it was broken up into two sentences.

*We have restructured the sentence into two sentences.*

4. Page 3, Figure 1: What are the small black arrows drawn between the volatility bins? They are not mentioned in the text or the caption and resemble a reverse aging scheme. I recommend explaining what is meant by the arrows or removing them.

*The black arrows represent the partitioning into the aerosol phase of a species when the aerosol loading increases to an order of magnitude above the species' volatility. We have added an explanation to the figure caption.*

5. Page 5: The organization between the "Dynamic Model" and "Results and Discussion" section could be clearer. Specifically, I would prefer to see a subsection within the "Dynamic Model" section (which could possibly be renamed "Methods") describing the experimental conditions of both the CLOUD and CMU experiments (i.e. moving the first paragraph from "Results and Discussion" here and adding a brief description of the CLOUD experiments). It will be helpful to the reader to have both experiments described in the same section for later reference.

*We have followed this excellent suggestion in our revisions.*

6. Page 5, line 21: I believe "low volatility" is meant in in place of "semi-volatile": the volatility bin that is referenced is 10-1 $\mu$g m$^{-3}$ and the corresponding abbreviation in parenthesis is "LVOC".

*Yes, this is now corrected.*

7. Page 6, section 2.2: It was unclear if the volatility distribution used was taken directly from Troestl et al. (2016) or if the authors performed the fit themselves. Furthermore, was the temperature correction done by Troestl et al. (2016) or the authors?

*The volatility distribution used was taken directly from Tröstl et al. (2016), and we conducted the temperature correction. We have modified the text to make this clear.*

8. Page 6: lines 31-32: I found the phrasing of this sentence awkward and had to reread it several times to understand it.

*We have changed the sentence structure to make it clearer.*

9. Page 8: equation (5): What are $v_{i,p}$ and $B_{i,p}$? These are not defined in the subsequent paragraph. What value is used for the accommodation coefficient $\alpha_{i,p}$?

*$v_{i,p}$ is the velocity of a molecule and $B_{i,p}$ is the Fuchs correction factor from the kinetic regime. We have added these descriptions. The value of the mass accommodation coefficient is 1.*

10. Page 9, line 4: I believe "(E)LVOC vapor wall loss" is meant in place of "(E)LVOC particle wall loss."

*Yes, we have changed this.*

11. Page 9, Figure 3: It is unclear how the light blue area, the Precursor Yield Potential, is calculated. The caption states that it is unreacted precursor, but line 7 of the following page states it is "the concentration of organics that have not yet been formed by ozonolysis." How is this calculated?

*The precursor yield potential is the (E)LVOC yield from $\alpha$-pinene ozonolysis multiplied by the remaining unreacted $\alpha$-pinene in the chamber. We have changed the figure caption to make this clearer.*

12. Page 11, line 9: What is a "perfect correction for the deposited particle mass"? Is this related to the lower or upper limit assumption (i.e., Loza et al., 2012)?

*This is not related to the lower or upper limit assumption. Particles deposited on chamber walls are difficult to sample and correct for. "Perfect correction" assumes that all deposited mass is accounted for in the total aerosol mass. We have modified the text to make this clear.*

13. Page 15, lines 24-15: What are the "constant HOM experiment" and the "increasing HOM experiment"'? These were not described previously.

*These experiments are described in-depth in Tröstl et al., 2016. We have added a reference to the descriptions in the text.*

14. Page 18, line 10: Please describe more explicitly what the base case is. Does the high (E)LVOC simulation include the additional LVOC concentration required to explain the observed growth rates in CLOUD?

*The base case is the high LVOC simulation, which includes the additional LVOC concentration that explains the CLOUD growth rates. We have made the text clearer.*

Referee 2:

1) The manuscript is motivated by the fact that yields of extremely low-volatile products in the CLOUD experiment are apparently much higher than in previous smog-chamber experiments. How large are the uncertainties in the assessed volatilities? Are the saturation vapor pressures of the $\alpha$-pinene oxidation products based on their O:C ratios and the SIMPOL model, as presented by Tröstl et al. (2016)? The recent study "$\alpha$-pinene autoxidation products may not have extremely low saturation vapor pressures despite high O:C ratios" by Kurtén et al. (2016) shows that group-contribution methods such as SIMPOL very likely underestimate the volatility of $\alpha$-pinene autoxidation products. SIMPOL predictions for the saturation vapor pressures of species studied in the work of Kurtén et al. are generally lower compared to other methods that are more suitable for describing the chemical interactions of autoxidation systems.

How would the comparison of the CLOUD and smog-chamber experiments look like if the saturation vapor pressures of the species of the CLOUD chamber were not as low as assumed by Tröstl et al.?

*It is important to realize that the volatilities in Tröstl et al. are heavily constrained by the **observed** growth rates. In that work (which is of course separately peer reviewed and published and not the subject of this discussion) the initial estimation of the volatility distribution is indeed based on SIMPOL-type estimations (and assumed structures consistent with the observed molecular composition and putative auto-oxidation mechanism). However, the predicted growth rates fall below observations by more than a factor of 2.5 over 3 hours, as shown for example in Extended Data Figure 6 of Tröstl et al. In order to balance the observed mass flux to the particles, the yields in the Tröstl et al. simulations over the LVOC range were increased significantly above the "sulfuric acid equivalent" values directly observed by the nitrate chemical ionization mass spectrometer by applying a self-consistent assumption that the nitrate clustering efficiency of LVOC vapors decreases with increasing volatility. In that regard the ultimate volatility distribution in Troöstl et al., is only marginally sensitive to the assumed volatility distribution of the directly observed produces, because the yields were scaled to match the (10 nm) growth rates in any event. Furthermore, the vapor saturation ratios in the simulations (Extended Data Figure 7 in Tröstl et al.) are for the most part orders of magnitude over 1, meaning that the exact saturation ratio makes very little difference other than at the smallest sizes. Figure 7 of our manuscript provides the same information for these simulations, and again during substantial SOA formation the saturation ratios are well above 100. If high volatility of observed species were to slow down the growth rate in CLOUD, the simulations would have needed to invoke even higher yields to explain the observed growth rates. This would cause an even greater mass-balance problem for the chamber data, which is the subject of this paper.*

2) When modeling the particle size evolution at the conditions of the CLOUD experiment, the role of ions and ion-ion recombination in the growth of the particle population is not addressed at all. Kirkby et al. (2016) state that the new-particle formation (NPF) events generated in the CLOUD chamber, from which the experimental growth rate (GR) data used in the present manuscript (as

well as in the study by Tröstl et al. (2016)) are extracted, are in practice driven by ions when the ion clearing field is off. The nucleation rates presented by Kirkby et al. imply that almost all particles that appear at dp = 1.7 nm at a HOM concentration of around $10^7$ cm$^{-3}$ are either positively or negatively charged, and "ion-ion recombination progressively neutralizes the charged particles as they grow".

I assume that at least the "constant HOM" data is from the CLOUD ion runs, as I don't see how the HOM concentration could otherwise be kept constant in the experiment while observing a NPF event. If this is so, the experimental growth rates determined from the time evolution of size-classified particle concentrations should be influenced by

-Recombination of particles of opposite polarities to form larger, electrically neutral particles.

-Possible enhancement in the condensation rate of molecules onto particles due to electrostatic interactions (charge enhancements for different compounds in the CLOUD chamber have been assessed by Lehtipalo et al. (2016)).

*This is a comment directed more at Kirkby et al. and Tröstl et al. than this manuscript, but we shall do our best to address it. First, there is no evidence for a large charge effect in the growth rate data presented by Troštl et al., as shown in Figure 1d of that paper. Second, it is straightforward to carry out a "constant HOM" nucleation experiment because the dominant sink of condensible vapors in CLOUD is the chamber wall, and so when the production rate is held constant the steady-state HOM concentration establishes itself with a 10-minute timescale (the wall-loss time constant). Third, the runs modeled in Tröstl et al. had fairly high monoterpene levels (2.4 μg m$^{-3}$) where even the 1 nm physical diameter neutral fraction is 80% of the formation rate. Fourth, there is really no telling what charge regime the CMU chamber experiments exist in; they are conducted in a non-conductive Teflon chamber under galactic cosmic ray conditions. Furthermore, for the cluster growth mechanism described in Lehtipalo et al to be important to growth rates, the mass concentration of small clusters has to be large compared to the mass concentration of condensible monomers. That situation can obtain when new-particle formation and growth are kinetically limited, which is the case for sulfuric acid + dimethyl amine, where dimers and trimers wind up containing a large fraction of the available sulfuric acid. This is far from the case for the organics. Nucleation is strongly sub-kinetic, which is plainly evident in the two order of magnitude charge enhancement at low concentrations. Further, the lack of a charge enhancement to growth under those same conditions confirms that cluster coagulation has little influence on the growth rates.*

3) Even when ions are not involved, coagulation of molecular clusters and small particles may also play a role in the evolution of the particle distribution. In the CLOUD experiment, this was previously addressed for NPF from sulfuric acid and bases by Lehtipalo et al. (2016), who concluded that nano-particle growth can be assisted by cluster coagulation in the presence of a strong stabilizing compound. While Lehtipalo et al. studied different chemical compounds than used here, the conclusion on the role of coagulation is general: the less small particles evaporate, the more their concentrations increase and thus the more significant the effect of coagulation becomes. This

should be relevant also for NPF from oxidized organics, if ELVOCs are capable of forming very stable small clusters -can the authors assess the effect of coagulation at the studied conditions?

*As we described in response to the previous comment, this mechanism can not be important for inefficient nucleation. Cluster enhancement to growth occurs when the clusters are extremely stable and thus compete with monomers in terms of concentration and collision rates with the small particles. Lehtipalo et al. were discussing dimethyl amine + sulfuric acid, where the particle formation is almost kinetic, and thus a large fraction of the sulfuric acid is bound up in dimers stabilized by the DMA. The organic conditions here are far from this case - the smallest clusters are evidently quite unstable, as the nucleation rates are at least $10^4$ below the kinetic collision rate and there is a large charge enhancement of about 100. Clusters under these conditions simply can not compete with condensation from the gas phase.*

4) In the CLOUD experiment comparisons, the particle growth model is applied to particle sizes down to around one nanometer. However, the validity of a single-particle model at very small nano-particle sizes is questionable. The model is "deterministic" in the way that it assumes that all particles of the same size grow at the same rate. It can't thus describe the fact that while the very smallest particle sizes may be unstable against evaporation so that the evaporation rate of molecules from the particle exceeds the collision rate of molecules, these unstable particles can still grow and generate a flux of particles towards larger sizes (this is the definition of nucleation). Instead, the used model requires the condensation rate to exceed the evaporation rate for all particle sizes in order for the particles to grow, even if this might not be the case in reality. It appears that the authors are trying to simulate the initial NPF process with a model that might not even in principle be capable of describing the phenomenon. Is it reasonable to start modeling the growth from the size of dp $\approx$ 1 nm, which corresponds to approximately a molecule or two? Is the growth rate of a single molecule even a reasonable concept?

*The reviewer is describing the nucleation process, as stated. Evaporation is by definition important below the "critical cluster" size. It is a direct consequence of having a process on the reactant side of the free-energy maximum along the reaction coordinate. The growth model here does include evaporation but as the reviewer notes we are assuming that the free energy maxima (this is a heterogeneous system and the small clusters almost certainly have a distribution of compositions) are for the most part smaller than the smallest modeled sizes. However, the consequences of opening up "evaporation to nothing" would be to change the nucleation rate, not the growth rates. We are not attempting to model J. Most importantly, the central problem for this paper is the high growth rate observed in CLOUD between 3 and 30 nm, where the issues raised by the reviewer are moot.*

5) Further, the evaporation of molecules from particles is described by the Kelvin approach throughout the modeled size range, and it is concluded that the Kelvin effect is important especially for the smallest sizes. On the other hand, the Kelvin approximation is based on classical macroscopic droplet thermodynamics, and is therefore not expected to be valid for small sizes of a few

molecules, which instead need to be treated with more sophisticated, atomic-level methods (e.g. Merikanto et al., 2007). Evaporation rates based on the Kelvin approach are thus likely to be highly uncertain; how does this uncertainty affect the results?

*The phenomenology will not change. We are not attempting to apply an observed surface tension to determine a "Kelvin diameter" but simply using the bulk Kelvin term to incorporate the phenomenon in a system with dozens of observed participating species with largely unknown physical properties. Whether it is arrived at via quantum calculations or by extrapolation from the bulk, it still holds that for the most part molecules that are less solvated will have fewer stabilizing interactions with other molecules and thus a higher free energy. We do not even know the molecular structures of the ELVOCs that are driving this process, and so it seems that high-level quantum calculations would not add accuracy to the model.*

Similarly, can condensed-phase oligomerization processes be expected to be similar for larger, macroscopic particles and for the smallest modeled sizes consisting of only a few molecules? Can the smallest sizes even be considered particle phase?

*Who knows? We are presenting a phenomenology that includes reactive uptake for a volatile monomer. In our model the monomer will react with \*any\* material in the particles with a single rate constant, so that gamma $\rightarrow$ 1 as the particles exceed 5 nm or so. We include a vapor pressure and again the Kelvin term generates the phenomenon that the monomer has a higher chance of evaporating from the smallest clusters and/or particles.*

6) The contribution of different VOC species to particle growth is obtained by fitting the model to experimental growth rates deduced from the "appearance times" of different particle sizes. Recent studies suggest that for the smallest, sub-3 nm sizes, the experimental appearance time-GR method can give distorted results, namely the apparent growth rates can be too high (Olenius et al., 2014; Kontkanen et al., 2016). Would the VBS fit change if the GRs of sub-3 nm sizes were lower?

*Of course. However, this is not so much a VBS fit as a model reproduction of the growth. If the "true" GR of the smallest particles were even lower, then a combination of a higher vapor pressure via either a higher C\* or a higher Kelvin diameter and/or somewhat lower ELVOC concentrations would be called for. None of this would substantially change the conclusion that effectively all of these organics are condensing by the time particles reach 10 nm, and if the chemistry is identical in CLOUD and SOA smog chambers, this same high condensation efficiency would occur in the smog-chamber studies. This in turn would imply very high SOA mass yields. That and not the CLOUD results is the actual subject of this study.*

Minor comments:

1) In the introduction, the expression "yield" is used in the context of both SOA mass yields and (E)LVOC yields; for easier reading, it would be good to better clarify which quantity is addressed when discussing the yields.

*"SOA mass yields" or "mass yields" is a general term that encompasses "(E)LVOC mass yields". In the areas where we discuss more narrow ranges, we either specify (E)LVOCs or the volatility range. We have added clarifications where this may be unclear.*

2) Page 12, lines 4-5: Please explain CSTR mode and batch mode, and also reformulate the expression "significantly higher than the (older) data" which is somewhat unclear.

*We have added these explanations.*

3) Page 13, line 5: The expression "For this simulation the suspended condensation sink was a few minutes" is not all clear ; please clarify what is meant by a sink being a few minutes.

*We mean "condensation sink timescale", and have changed the text accordingly.*

4) Page 13, line 35: When discussing the nucleation rate data from Kirkby et al., are the authors referring to the neutral or ion-induced rates?

*We are referring to the neutral rates.*

5) Page 14, lines 1-5: "The data in the log-log plot have a slope of 2, indicating that the nucleation rate is a second order reaction with respect to the ELVOC concentration": generally, the slope does not indicate something about reactions (e.g. Ehrhart and Curtius, 2013). The derived "nucleation rate constant" doesnt have an actual physical meaning, as the rate is not affected only by the vapor concentration (and cluster evaporation), but also by e.g. losses of the formed molecular clusters onto chamber walls and surfaces of larger particles. The slope of the nucleation rate, regardless of what all the dynamic processes affecting the nucleation are, also varies with the absolute vapor concentration (as well as other factors affecting the nucleation process), and therefore applying the "nucleation rate constant" to a completely different experimental set-up is questionable. Large sinks, i.a. large suspended particles, can efficiently suppress nucleation; could this be related to the fact that particle formation was not observed in the smog-chamber experiments? Moreover, how was the effect of temperature on the nucleation rate taken into account when assessing rates for a smog-chamber?

*Certainly, the conditions of the chamber other than the ELVOC concentrations can have an impact on the nucleation rates. However, we are not attempting to produce an exact solution to*

*nucleation here. We pose this hypothetical given what we know from CLOUD to demonstrate the strong nucleation that should have occurred in the CMU chamber given all else being equal. Whether other factors can explain the lack of nucleation is up to further studies to determine.*

6) Figure 8: Why are the growth rates of specific particle sizes much higher in the "increasing HOM" case? Are the absolute vapor-phase HOM concentrations similar for the "constant HOM" and "rising HOM" experiments? Why is the particle size range different for the two experiments? Same x- and y-axis limits would make the figure easier to read. Why are some particle appearance times negative in panel c? How is t = 0 defined? It is stated that "the experimentally determined growth rate at 10 nm matches the model", but no experimental growth rates are presented; please show also GRs deduced from the CLOUD observations in Figure 8.

*The growth rates are different because the "increasing HOM" case had higher levels of (E)LVOCs than the "constant HOM" case. The particles therefore grow to a larger size. As for the reviewer's comments on the plots, these plots follow those published in Tröstl et al. 2016.*

7) Are the presented particle diameters mass or mobility diameters?

*The particle diameters are mobility diameters.*

Technical comments:

-Page 11, line 10: Should "$\Delta\alpha$-pinene/$C_{OA}$" be "$C_{OA}/\Delta\alpha$-pinene"?

*Yes, this has been corrected.*

-Page 12, line 4: Remove "also" from the sentence "In Figure 5 we also also show..."

*Corrected*

-Figure 1: Explain the markings with alphas on the right-hand side of the figure frame. Also state that the figure (minus the offset bar and the green curve) is taken from Presto and Donahue (2006).

*We have added explanations.*

-Figure 3, figure caption: Expressions "Experiment 1 monodisperse (a) and polydisperse (b) results. Experiment 2 monodisperse (c) and polydisperse (d) results." are somewhat clumsy; please replace them with proper sentences.

*We have replaced the sentences.*

-Figure 6: The upper limit of the y-axis could be set to 1, as the white space above the value of 1 is unnecessary.

*Changed*

-Figure 7 a and b: A legend would be helpful. Also, an extra occurrence of the word "experiments" should be removed from the figure caption: "Vapor saturation ratios for smog-chamber experiments experiments", and "dark gray" should be "dark grey". The unit of the nucleation rate on the y-axis label should be particles $cm^{-3}$ $s^{-1}$, not molec. $cm^3$ $s^1$.

*Corrected*

-Figure 8: A legend would be good here, too. In the caption, "oligomerizeration" should be "oligomerization".

*Corrected*

-Figure 10: In the caption, "blue triangles" should presumably be just "blue".

*Corrected*

-Figure 12: I don't see the discussion starting with "At a low CS ratio, fewer seeds..." belonging to a figure caption (and in any case, essentially the same information is already included in the main text, so there is no need to repeat it). Similarly for Figure 13: the explanation starting with "Given a seed concentration..." shouldn't be a part of a caption.

*We prefer figure captions that allow readers to peruse the figures and understand the paper.*

-For all figures that present same quantities for different experimental set-ups or simulation cases in separate panels (Figures 3-5, 7-9, and 11-13), adding titles to the panels would make the figures easier to read.

*We have added legends.*

-There are quite many separate figures in the manuscript. Could some of the figures be combined together as panels of a larger figure (e.g. Figures 4 and 5), or merged into the same panel (e.g. Figure 5 and 10)?

*It is a long subject - in our opinion if we merged figures they would become too complicated and thus hard to follow.*

List of relevant changes. Line numbers correspond to the latexdiff version. There are additional changes noted in the latexdiff file.

1. page 7, line 1: Added subsection title "Experimental Data" before the paragraph "In a typical..."

2. page 7, lines 12-23: Added paragraphs to the "Experimental Data" section.

   " The CLOUD chamber experiments are described in detail in Tröstl et al. (2016); we give a brief summary here. Two types of $\alpha$-pinene ozonolysis experiments were conducted: those with increasing HOM concentrations and those with steady-state HOM concentrations. Using the constraints on the aerosol growth rate from both types of experiments, they derived volatility-distributed mass yields of the products that reproduced particle sizes.

   To compare the CLOUD (E)LVOC mass yields with smog-chamber experiments, we simulate data from two experiments described by Pathak et al. (2007b), both of which had relatively high initial seed surface area and thus should have had low (E)LVOC vapor wall loss and rapid equilibration. Both experiments were conducted near room temperature, common in many other smog-chamber experiments. Experiment 1 was conducted with 17 ppb $\alpha$-pinene, a constant 250 ppb $O_3$, and 12000 cm$^{-3}$ ammonium-sulfate seeds. Experiment 2 was conducted with 38.3 ppb $\alpha$-pinene, a constant 250 ppb $O_3$, and 6000 cm$^{-3}$ ammonium-sulfate seeds. The SMPS data for these experiments show clear volume maxima after SOA condensation as well as periods where particle wall losses clearly dominate; these are essential to constrain the model. The data also show a steady decline in total particle number with no sign of nucleation after the onset of $\alpha$-pinene ozonolysis."

3. page 19, line 10: Changed "Other dynamical effects also remain possible explanations." to "Other dynamic effects also remain possible explanations, though they require future experiments to constrain."

4. page 1, line 5: Changed "dynamical" to "dynamic"

5. page 4, line 12: Changed "Dynamical" to "Dynamic"

6. page 5, line 28: Changed "dynamical" to "dynamic"

7. page 12, line 17: Changed "dynamical" to "dynamic"

8. page 12, line 18: Changed "dynamical" to "dynamic"

9. page 13, line 3: Changed "dynamical" to "dynamic"

10. page 13, line 14: Changed "dynamical" to "dynamic"

11. page 14, line 1: Changed "dynamical" to "dynamic"

12. page 14, line 9: Changed "dynamical" to "dynamic"

13. page 14, line 11: Changed "dynamical" to "dynamic"

14. page 19, line 10: Changed "dynamical" to "dynamic"

15. page 23, line 9: Changed "dynamical" to "dynamic"

16. Figure 3 caption: Changed "Dynamical" to "Dynamic"

17. page 1, lines 12-13: Added the sentence "The oligomerization simulations are a closer match to the CMU experiments than the base-case simulations, though they over-predict the observations somewhat."

18. page 1, line 24 - page 2, line 2: Changed "...theory and furthermore "Odum plots"..." to "...theory. Furthermore, "Odum plots"..."

19. Figure 1 caption: Added a sentence, "Black arrows denote the complete partitioning of lower volatility products when the total aerosol mass increases by an order of magnitude." before the sentence "The gap between the black..." Added sentences "The $\alpha$'s on the right side denote the yields derived for each subsequent volatility bin." and "Adapted from Presto and Donahue (2006)."

20. Figure 3 caption: Replaced "Experiment 1 monodisperse..." and "Experiment 2 monodisperse..." with "Figures **(a)** and **(b)** are the monodisperse and polydisperse results for Experiment 1, respectively. Figures **(c)** and **(d)** are the monodisperse and polydisperse results for Experiment 2, respectively."

21. page 6, line 1: Changed section title "The Dynamic Model" to "Methods".

22. page 9, lines 2-9: Deleted first paragraph "To compare the CLOUD ... after the onset $\alpha$-pinene ozonolysis."

23. page 6, line 6: Changed "semi-volatile" to "low-volatile"

24. page 8, line 32: Added to the sentence "By applying the Clausius-Clapeyron equation..." to become "By applying the Clausius-Clapeyron equation to the volatility distribution from Tröstl et al. ..."

25. page 7, line 30-page 8, line 2: Made a new paragraph, and changed the sentence structure. "To compare the CLOUD results with CMU smog-chamber data, we need to correct for different experimental temperatures because volatility depends strongly on temperature. The CMU smog-chamber experiments were conducted near room temperature; however, the CLOUD chamber experiments were conducted at 278 K."

26. page 8, lines 26-27: Added "$v_{i,p}$ is the velocity, $B_{i,p}$ is the Fuchs correction factor from the kinetic regime" after "...$\alpha_{i,p}$ is the accommodation coefficient,"

27. page 10, line 3: "particle wall loss" is changed to "vapor wall loss ". (This paragraph was moved to the Methods section)

28. page 9, Figure 3 caption: Changed to " The simulations describe five different reservoirs: potential product from unreacted $\alpha$-pinene, vapors, suspended particles, deposited particles, and sorption to teflon, as shown in the legend. Figures **(a)** and **(b)** are the monodisperse and polydisperse results for Experiment 1, respectively. Figures **(c)** and **(d)** are the monodisperse and polydisperse results for Experiment 2, respectively. The monodisperse model uses a "condensation sink diameter" to approximate the rate that organics condense onto particles. This serves as a good proxy for a polydisperse model that accounts for the different condensation sinks for a polydisperse seed distribution."

29. page 12, line 10: Changed "In this case we assume a perfect correction for the deposited particle mass..." to "In this case we assume that all deposited particle mass is accounted for..."

30. page 16, lines 30-31: Added sentence after "increasing HOM experiment" – "Descriptions of these experiments are detailed in Tröstl et al. (2016)."

31. page 19, line 17: changed "(E)LVOC" to "LVOC".

32. page 1, line 26: Changed "mass yield" to "SOA mass yield"

33. page 4, line 19: Changed to "a high (E)LVOC mass yield"

34. page 4, lines 22-23: Changed "observed mass yield" to "empirical (E)LVOC mass yield"

35. page 5, line 16: Changed "mass yield" to "SOA mass yield"

36. page 13, lines 9-11: Added the sentence "A CSTR, or continuous flow stirred-tank reactor, has a constant flow of reactants into the chamber and product mixture out of the chamber, while a batch reactor does not." after "...comparison should be made with care."

37. page 13, line 7: removed "the (older) data" and added "Pathak et al. (2007) and Presto and Donahue (2006)."

38. page 14, line 10: Changed "condensation sink" to "condensation sink timescale".

39. page 11, line 10: Changed "$\Delta\alpha$-pinene$/C_{\mathrm{OA}}$ to "$C_{\mathrm{OA}}/\Delta\alpha$-pinene

40. page 13, line 6: removed an 'also' from the first sentence

41. Figure 7 caption: Removed an instance of "experiments" from "...smog-chamber experiments experiments".

42. Figure 8 caption: Corrected "Oligomerizeration" to "Oligomerization".

43. Figure 10 caption: Changed "blue triangles" to "blue".

[revised manuscript text omitted]

---

## Author Response (AR2)

1) Nanoparticle coagulation: It is stated that coagulation of small particles cannot be important for the CLOUD growth rate observations for organics, as opposed to the results of Lehtipalo et al. for acid-base mixtures. The reason for bringing up the Lehtipalo et al. work was that the study discusses the significance of self-coagulation for growth also with respect to the saturation vapor pressure of the clustering vapor. Lehtipalo et al. conclude that the observed growth rates can be explained by assuming $P^{sat} = 0$, at which coagulation significantly contributes to the growth. Test model runs, shown in Fig. S6 in that paper, show however that essentially the same growth rates are reached already at $P^{sat} = 5 \times 10^{-9}$ Pa as self-coagulation processes become important when cluster evaporation is very low. The lowest saturation vapor pressures assumed in the current manuscript are of the order of around $10^{-14}$ (ELVOCs)...$10^{-6}$ (LVOCs) Pa, but in their replies the authors state that that cluster self-coagulation is negligible. If self-coagulation is important at $P^{sat} = 5 \times 10^{-9}$ Pa, how can it not be important below it? While it may be that coagulation is not significant for organics in the actual CLOUD experiment, it must be significant in particle modeling with these model parameters - $P^{sat}$ values used for the ELVOCs lead to elevated concentrations of stable small particles. (A small sidenote about the model assumptions: The statement about critical sizes being smaller than the smallest modeled size, which is the size of one molecule, is hardly justifiable - or if they are approximately the size of a monomer unit, clustering should be in practice kinetically controlled which brings us back to the coagulation question.)

*Self coagulation is certainly important under some conditions, and Lehtipalo et al. demonstrated its effects on growth rate for particles < 3 nm when the production rate of low-volatility compounds was very high. However, just because there are some compounds present with $P^{sat} < 5 \times 10^{-9}$ Pa does not mean that self coagulation is necessarily important - this also depends on the overall production rate and the ratio of monomers to small clusters, etc. For the organics, the ELVOCs with such low vapor pressures were only present at sufficient levels to drive growth rates of 1 nm/hr or so. Specifically, the concentration of ELVOCs in CLOUD was on the order of $10^5\,cm^{-3}$ , far lower than the concentrations shown in Lehtipalo et al. Most importantly, we are ultimately interested in how the observed growth rates affect the mass production rate of condensible vapors; therefore, we require to match those growth rates. In that context, what occurs between 1 and 3 nm is not overly relevant; what drives growth at 10 nm is critical. So, for the simulations here, we focus on matching growth rates at 10 nm, where the contribution of coagulation to growth is greatly reduced. At those sizes, condensation is the main driver behind particle growth. However, we recognize the importance of eventually capturing the coagulation process, and we have added this to the manuscript as an area that requires further modeling.*

2) Effects of ions and recombination: The observed growth rates are indeed mostly of the same order of magnitude for runs with and without ions, but it is not direct evidence about the particle formation mechanism being the same. The main point is that if there are

elevated concentrations of small ion particles as the data show, recombination processes are likely to lead to faster formation of somewhat larger, neutral particles. Recombination rate constants are expected to be 3-4 orders of magnitude higher than vapor condensation rate constants, which makes the collision frequencies of small charged clusters of opposite polarities comparable to those of vapor molecules.

To summarize, it is clear that the condensation modeling is a valid approach for larger sizes which are the main topic of the study. It may, however, be useful to acknowledge that dynamic processes other than simple vapor condensation may contribute to the observed high growth rates of especially the smallest particles. In this case, the least volatile species don't need to be as low-volatile as assumed, and/or they don't need to exist at as high concentrations as assumed. This remark is not meant as criticism at the present modeling study, but as an attempt to suggest additional reasons to explain the rapid appearance of subsequent particle size classes in the CLOUD chamber.

*Indeed the effect of charged clusters can enhance the growth of particles, and this phenomenon should be considered for a model aiming to predict growth rates for sub-3 nm particles. At the sizes that we focus on here, ion recombination is unlikely to have a large effect, although the exact degree is not yet determined. Figure 3 in Lehtipalo et al. shows the enhancement to the growth rate decreasing from 1.5 nm to 2 nm, and will likely continue to decrease with increasing particle size. In this work, we aimed to match particle growth at 10 nm, a point where ion effects should be small. If in fact the recombination effect remains substantial at larger sizes, it would exacerbate the overprediction that we encountered when modeling chamber experiments. This is not to say that ion recombination is not important, only that it does not appear to offer a solution to the apparent mis-match between CLOUD and smog-chamber observations. Still, the phenomenon is worth acknowledging, and we have added that to the manuscript.*

Minor comment: Constant HOM and increasing HOM: The explanation about the HOM reaching steady state in 10 minutes is not evident in the CLOUD data - it seems to take much longer according to the figures in Kirkby et al. and Tröstl et al., and this is the reason for asking the original question.

The time scale can of course be assessed from the loss rate assuming a constant source - although the HOM source isn't exactly constant as the alpha-pinene concentration increases- and negligible losses on particles: reaching e.g. 90% of the steady-state value should take about a half an hour (during which particles should already be appearing).

But the "constant HOM" case in Fig. 3 in Tröstl et al. does not show any increase during the first 10 minutes, or at any times, and the only experiment shown where [HOM] does not increase is the ion run (Fig. S8 in Tröstl et al.)

I understand that this point is more relevant to the Tröstl et al. work, but as I could not

find in that paper any explanation for the "constant" and "increasing" HOM cases, it is appropriate to clarify them in the current manuscript since these cases are simulated and discussed here.

*The reviewer appears to be confused between two different timescales in CLOUD. The HOMs do indeed reach a steady state in roughly 10 minutes (e-folding), which is determined by their wall-loss time constant. However, non-sticky gases such as $\alpha$-pinene reach a steady state over a much longer timescale, roughly 3-hours, which is the flushing time constant of the chamber. For the "constant HOM" case the $\alpha$-pinene was at steady state and nucleation was initiated by turning off the clearing field. Figure S8 from Tröstl et al. shows an initial increase in HOM concentration because $\alpha$-pinene levels are increasing, and subsequently settles to a constant HOM concentration when $\alpha$-pinene is held constant. We agree that it would help to clarify the different experiments in the manuscript and have added descriptions.*

1. page 7, lines 5-7: Added clarification for the CLOUD experiments.

2. page 18, lines 15-17: Added explanations for other potential growth rate factors.

[revised manuscript text omitted]